# DEFAME: Dynamic Evidence-based FAct-checking with Multimodal Experts

**Tobias Braun** [* 1]  **Mark Rothermel** [* 1]  **Marcus Rohrbach** [1]  **Anna Rohrbach** [1]

## Abstract

The proliferation of disinformation demands reliable and scalable fact-checking solutions. We present **D**ynamic **E**vidence-based **FA**ct-checking with **M**ultimodal **E**xperts (DEFAME), a modular, zero-shot MLLM pipeline for open-domain, text-image claim verification. DEFAME operates in a six-stage process, dynamically selecting the tools and search depth to extract and evaluate textual and visual evidence. Unlike prior approaches that are text-only, lack explainability, or rely solely on parametric knowledge, DEFAME performs end-to-end verification, accounting for images in claims *and* evidence while generating structured, multimodal reports. Evaluation on the popular benchmarks VERITE, AVERITEC, and MOCHEG shows that DEFAME surpasses all previous methods, establishing itself as the new general state-of-the-art fact-checking system for uni- and multimodal fact-checking. Moreover, we introduce a new multimodal benchmark, CLAIM-REVIEW2024+, featuring claims after the knowledge cutoff of GPT-4O, avoiding data leakage. Here, DEFAME drastically outperforms the GPT-4O baselines, showing temporal generalizability and the potential for real-time fact-checking[2].

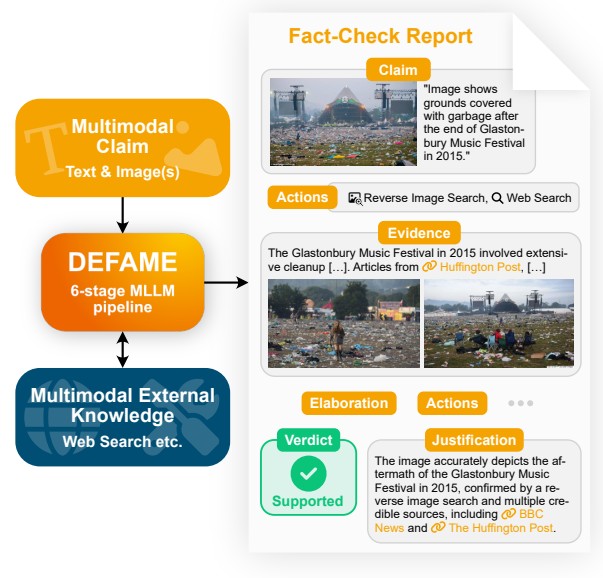

Figure 1: DEFAME in a nutshell: It fact-checks multimodal claims using multimodal evidence and returns a detailed, human-friendly report document.

## 1. Introduction

In recent years, misinformation has been growing in scale and quality (Chen & Shu, 2024) beyond human capacity to fact-check. "Fake news" has evolved from a lighthearted term into a serious global threat (World Economic Forum, 2024). Driven by higher engagement rates on social media and the increase in AI use, misinformation spreads faster, reaches a broader audience, and causes greater harm (Li & Xie, 2020; Zannettou et al., 2018; Wang et al., 2020; Chen & Shu, 2024). Humans perceive multimodal information as more credible (Newman et al., 2012; Hameleers et al., 2020), often interpreting visuals as "evidence" (Greifeneder et al., 2021), making it particularly persuasive. Approximately 80% of the claims checked by professional fact-checkers are multimodal (Dufour et al., 2024)—signifying a strong priority for checking multimodal content.

Unfortunately, Automated Fact-Checking (AFC) systems are mostly text-only (Dmonte et al., 2024; Vykopal et al., 2024). Only a few works venture into visual claim verification (Khaliq et al., 2024; Shao et al., 2023), but *none* can handle both multimodal claims and multimodal evidence at once. Most multimodal claim verification systems cannot retrieve the evidence needed to verify a claim (Fu et al., 2024; Vo-Hoang et al., 2024; Tang et al., 2024). The majority of AFC works focus on a specific aspect of AFC such as evidence retrieval (Cheung & Lam, 2023), evidence summarization (Chen et al., 2024), or evidence ranking (Tahmasebi

---

*Equal contribution  [1]Technical University of Darmstadt & hessian.AI, Germany. Correspondence to: Mark Rothermel <mark.rothermel@tu-darmstadt.de>.

*Proceedings of the 42nd International Conference on Machine Learning*, Vancouver, Canada. PMLR 267, 2025. Copyright 2025 by the author(s).

[2]We released the code and benchmark dataset publicly at: https://github.com/multimodal-ai-lab/DEFAME/tree/icml

et al., 2024). This specialization has created a scattered landscape where individual approaches address isolated aspects of a complex problem. AFC systems that target the overall task of fact-checking typically are text-only (Zhao et al., 2024; Li et al., 2024c), lack performance (Chen et al., 2024; Cao et al., 2023; Papadopoulos et al., 2024a), do not involve a flexible planning module (Khaliq et al., 2024; Tahmasebi et al., 2024), or are not explainable (Shao et al., 2023; Xu et al., 2024).

Therefore, as the main contribution of this paper, we introduce DEFAME, a straightforward end-to-end AFC framework, unifying the advancements in the field of AFC into one single system. As the first of its kind, it is able to natively process multimodal claims *and* evidence, the latter of which it retrieves dynamically as needed. DEFAME is designed with transparency in mind, imitating a human fact-checking process and returning a detailed fact-check report to the user (cf. Figure 1).

We empirically demonstrate DEFAME's effectiveness and generality by establishing new state-of-the-art results on three diverse and widely used benchmarks, surpassing even the most specialized methods. On AVERITEC (Schlichtkrull et al., 2024b), we improve accuracy from 65.6% to 70.5%; on MOCHEG (Yao et al., 2023), we achieve a +10.6% improvement in accuracy, and on VERITE (Papadopoulos et al., 2024b), we enhance True/False accuracy by +25.9%.

Additionally, we contribute a new benchmark, CLAIMRE-VIEW2024+, with claims that occurred after the knowledge cutoff of GPT-4O to mitigate the effects of data leakage. We show that DEFAME is superior to GPT-4O on these "unseen" statements. We show that the fact-checking reports generated by DEFAME are well-received by human evaluators, who prefer them over GPT-4O's outputs. Finally, we make our code and benchmark publicly available[3].

## 2. Related Work

Automating the task of fact-checking is a difficult problem, far from being solved (Akhtar et al., 2023b; Dmonte et al., 2024; Yao et al., 2023; Schlichtkrull et al., 2024a). Due to its complexity, Akhtar et al. (2023b) subdivide AFC into three stages: (1) claim detection & extraction, (2) evidence retrieval, and (3) verdict prediction. Most approaches narrow down their scope, either by focusing on a sub-task like summarization (Chen et al., 2024), justification generation (Atanasova et al., 2020), evidence retrieval (Samarinas et al., 2021), deepfake detection (Jia et al., 2024), out-of-context detection (Xu et al., 2024; Vo-Hoang et al., 2024), image contextualization (Tonglet et al., 2024), or by addressing

only a specific domain, e.g., charts (Akhtar et al., 2023a; 2024), social media (Wang et al., 2018), politics (Khaliq et al., 2024), or news (Xu et al., 2024; Shao et al., 2023). Others investigate the incorporation of new architectures like a knowledge graph (Cao et al., 2024) or address the problem of evidence ambiguity (Glockner et al., 2024). In contrast to all these methods, DEFAME combines the fragmented landscape of AFC work into *one end-to-end solution*—not restricted to only one modality, domain, or subtask. Table 1 compares DEFAME to prior work.

**Text-only fact-checking**: The vast majority of proposed AFC systems is purely text-based (Hassan et al., 2017; Thorne et al., 2018; Schlichtkrull et al., 2024b; Dmonte et al., 2024; Vykopal et al., 2024; Yang & Rocha, 2024; Zhao et al., 2024; Wang et al., 2024b; Li et al., 2024b; Pan et al., 2023; Schlichtkrull et al., 2024a; Cheung & Lam, 2023). The most popular benchmarks to evaluate these methods are LIAR (Wang, 2017), FEVER (Thorne et al., 2018), and AVERITEC (Schlichtkrull et al., 2024b), the latter of which mitigates important weaknesses of the previous ones, including reliance on artificial claims and evidence insufficiency. For this reason, we use AVERITEC in our evaluation. Most recently, FACTCHECK-BENCH (Wang et al., 2024a) was introduced to evaluate the factuality of entire LLM responses.

**Multimodal fact-checking**: Popular multimodal AFC benchmarks include works from Xu et al. (2024); Nielsen & McConville (2022); Zlatkova et al. (2019); Nakamura et al. (2020); Aneja et al. (2021); Yao et al. (2023); Jaiswal et al. (2017); Sabir et al. (2018); Müller-Budack et al. (2020); Luo et al. (2021a). Notably, MOCHEG (Yao et al., 2023) builds on real-world claims, requiring multimodal evidence retrieval, additionally incorporating the task of justification generation. Given these features, we use MOCHEG in our evaluation. A strong emphasis of multimodal AFC has been on Out-Of-Context (OOC) image detection, sometimes referred to as "cheapfake" detection. Popular evaluation benchmarks include NEWSCLIPPINGS (Luo et al., 2021b) and, more recently, VERITE (Papadopoulos et al., 2024b). In our experiments, we use VERITE because it improves over previous benchmarks by reducing unimodal bias and incorporating real-world samples.

Some multimodal AFC solutions utilize multimodal fusion (Papadopoulos et al., 2023; 2024a; Shao et al., 2023; Zeng et al., 2024b), leverage inter-, and cross-modal consistency (Abdelnabi et al., 2022; Fu et al., 2024), or apply conventional machine learning models (Papadopoulos et al., 2024a; Wang et al., 2018). A strong disadvantage of these systems is the inability to produce human-understandable explanations of the predictions. Furthermore, in stark contrast to DEFAME, they often rely on superficial pattern matching or lexical/visual similarity, ignoring factuality and

---

[3] https://github.com/multimodal-ai-lab/DEFAME/tree/icml

| Method | Multimodal claims | Multimodal evid. | Evidence retrieval | Multi-hop | Planning | Tools | Reasoning | Explainable | Training-free | Open-domain | Open source | Backbone |
|---|---|---|---|---|---|---|---|---|---|---|---|---|
| RAGAR (Khaliq et al., 2024) | ✓ | - | ✓ | ✓ | - | - | ✓ | ✓ | ✓ | - | - | MLLM |
| MMD-AGENT (Liu et al., 2025) | ✓ | - | ✓ | - | ✓ | - | ✓ | ◑ | ✓ | - | ✓ | MLLM |
| SNIFFER (Qi et al., 2024) | ✓ | - | ✓ | - | - | ✓ | ✓ | ✓ | - | - | ✓ | VLT & LLM |
| MMOOC-CHECKER (Xu et al., 2024) | ✓ | - | ✓ | - | - | - | - | - | - | - | - | VLT |
| VO-HOANG ET AL. (Vo-Hoang et al., 2024) | ✓ | - | - | - | - | - | - | - | - | ✓ | - | LLM & VLT |
| ZENG ET AL. (Zeng et al., 2024a) | ✓ | - | - | - | - | - | - | - | - | ✓ | - | MLLM |
| CHASMA (Papadopoulos et al., 2024b) | ✓ | - | - | - | - | - | - | - | - | ✓ | ✓ | CLIP & VLT |
| AITR (Papadopoulos et al., 2024a) | ✓ | - | - | - | - | - | - | - | - | ✓ | ✓ | VLT |
| HAMMER (Shao et al., 2023) | ✓ | - | - | - | - | - | - | - | - | - | ✓ | BERT & ViT |
| MULTIMD (Fu et al., 2024) | ✓ | - | - | - | - | - | - | - | - | - | - | VLT |
| LVLM4FV (Tahmasebi et al., 2024) | - | ✓ | ✓ | - | - | - | ✓ | ◑ | ✓ | ✓ | ✓ | (M)LLM |
| MOCHEG (Yao et al., 2023) | - | ✓ | ✓ | - | - | - | ◑ | - | - | ✓ | ✓ | VLT |
| METASUM (Chen et al., 2024) | - | ✓ | ◑ | - | - | - | - | ✓ | - | ✓ | - | VLT & LLM |
| M³D (Tang et al., 2024) | - | ✓ | - | - | - | - | - | - | - | - | - | VLT & GCN |
| CHARTBERT (Akhtar et al., 2023a) | - | ✓ | - | - | - | - | - | - | - | - | - | BERT |
| PACAR (Zhao et al., 2024) | - | - | ✓ | ✓ | ✓ | ✓ | ✓ | ✓ | ✓ | ✓ | - | LLM |
| HISS (Zhang & Gao, 2023) | - | - | ✓ | ✓ | - | - | ✓ | ✓ | ✓ | ✓ | - | LLM |
| PROGRAMFC (Pan et al., 2023) | - | - | ✓ | ◑ | ✓ | ✓ | ◑ | ◑ | ✓ | ✓ | ✓ | LLM |
| SELF-CHECKER (Li et al., 2024c) | - | - | ✓ | - | ✓ | - | ✓ | ◑ | ✓ | ✓ | - | LLM |
| FACTLLAMA (Cheung & Lam, 2023) | - | - | ✓ | - | - | - | - | - | - | ✓ | - | LLM |
| CFR (Sriram et al., 2024) | - | - | ✓ | - | - | - | - | - | - | ✓ | - | BERT & LLM |
| DEBERTA (Cao et al., 2023) | - | - | - | - | - | - | ✓ | ◑ | ✓ | ✓ | - | LLM |
| **DEFAME (Ours)** | ✓ | ✓ | ✓ | ✓ | ✓ | ✓ | ✓ | ✓ | ✓ | ✓ | ✓ | MLLM |

Table 1: Overview of the most relevant and published AFC systems. We consider a method "Explainable" if it offers substantial (✓) or some (◑) human-readable information explaining the decision. Methods with "VLT" backbones employ a model from the Vision Language Transformer family.

logic, raising questions about their robustness and actuality.

**(M)LLM-based fact-checking**: With the rise of (Multimodal) Large Language Models or (M)LLMs, the AFC community increasingly explored prompt-driven solutions (Yang & Rocha, 2024; Khaliq et al., 2024; Tahmasebi et al., 2024; Chen et al., 2024; Pan et al., 2023; Cheung & Lam, 2023). One of DEFAME's closest relatives, RAGAR (Khaliq et al., 2024), processes textual and visual claims but retrieves only textual evidence. Furthermore, DEFAME directly incorporates the claim image in its context while RAGAR converts it into a textual description, discarding critical visual information.

While there have been efforts to improve the performance of MLLM-based approaches through training on synthetic data (Zeng et al., 2024a), most still rely on the parametric knowledge of the MLLMs (Beigi et al., 2024; Shao et al., 2023; Li et al., 2024e; Geng et al., 2024), foregoing external evidence retrieval. This approach has three major drawbacks: (1) MLLM knowledge is static and fails on recent claims, as shown in this work; (2) predictions lack links to verifiable sources, reducing transparency; and (3) reliance on parametric knowledge increases hallucination risks, making such methods less reliable than Retrieval-Augmented Generation (RAG)-based approaches. In contrast, DEFAME uses internal knowledge mostly for commonsense reasoning, retrieving evidence dynamically through external tools. Unlike some prior work (Yang & Rocha, 2024; Tang et al., 2024), it does *not* rely on benchmark-provided gold evidence, reinforcing its adaptability to real-world misinformation.

## 3. DEFAME Approach.

Large Language Model (LLM) agents have become a powerful solution for commonsense reasoning, summarization, basic planning, and tool use (Davis, 2023; Li et al., 2024d; Zeng et al., 2023; Surís et al., 2023). DEFAME (see Figure 2) comprises a Multimodal LLM (MLLM), a suite of multimodal tools, and a structured fact-check report. Our proposed framework effectively operates as a dynamic, multi-step RAG system (Lewis et al., 2020), inspired by established fact-checking workflows (Moreno Gil et al., 2021). Each call to the MLLM includes the current state of the fact-checking report as contextual input, along with a task-specific description. This approach emulates a form of context awareness, guiding the MLLM to focus on pertinent information at each stage of the fact-checking process and allowing for more intricate, multi-hop reasoning and evidence retrieval. Nonetheless, LLM's limitations like hallucinations, knowledge cutoff, and stochastic outputs (Maynez et al., 2020; Li & Flanigan, 2024) necessitate careful management. Accordingly, we decompose the fact-checking process into six manageable stages, five of which are subject to MLLM prompting. The procedure mimics human

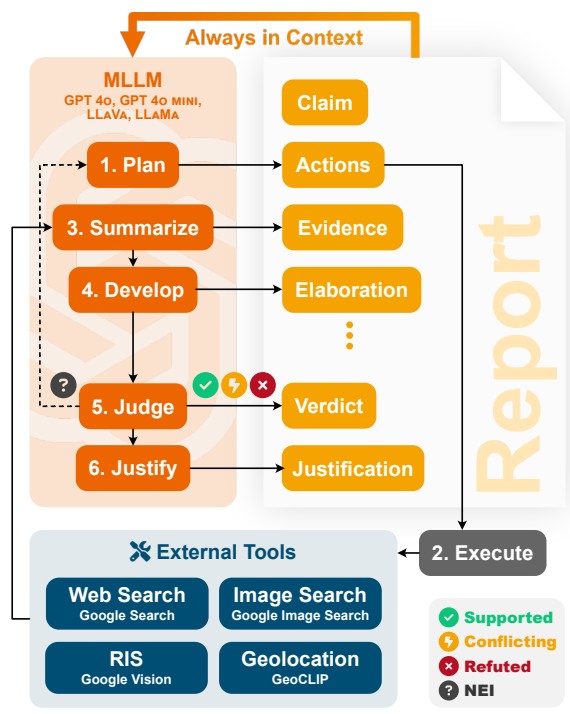

Figure 2: Overview of DEFAME's dynamic six-stage pipeline involving three main components: an MLLM, a fact-checking report, and external tools.

fact-checkers and is described in detail in the following.

**Stage 1: Plan Actions.** Upon receiving a claim, the MLLM is prompted to suggest a targeted action sequence to retrieve missing information. Since the action space is infinite for some tools (e.g., web search allows arbitrary queries), the "planner" aims to minimize actions and cost. To prevent redundancy, DEFAME tracks previously executed actions and adapts if it encounters a "dead end." In-context learning guides the module in deciding which of the specialized tools to invoke: Web Search, Image Search, Reverse Image Search (RIS), or Geolocation. While RIS and Geolocation handle image inputs, Web Search and Image Search operate on dynamically generated text queries.

**Stage 2: Execute Actions.** Given a set of actions, DEFAME invokes the corresponding tool:

1. **Web Search**: Given a textual search query, it leverages Google Search via the Serper API[4] to provide the top 3 relevant web pages matching the query. This gives DEFAME foundational access to the open web, enabling it to retrieve current or very domain-specific evidence that is not encoded in the MLLM's parameters.

2. **Image Search**: It applies Google Image Search to return up to 3 URLs of web pages containing images, diagrams, and infographics that match a given textual caption. These retrieved visuals can serve as evidence in the report and/or as inputs for DEFAME's Geolocation and RIS tools.

3. **Reverse Image Search (RIS)**: For a given image, it retrieves up to 3 URLs of web pages containing the same image, using the Google Vision API[5]. It serves as a real-time image knowledge base, enabling DEFAME to contextualize a given image with information such as image sources, authors, prior uses, and approximate dates——that a static trained MLLM, with fixed parametric knowledge, cannot offer.

4. **Geolocation** integrates GEOCLIP (Cepeda et al., 2023)—a specialized geolocation tool designed to estimate the most probable countries from which an image could originate. MLLMs lack the ability for this AFC-critical task (Roberts et al., 2024).

To prevent temporal leakage, all web-based tools restrict search results to sources published before the claim's release date (if known). Additionally, we exclude major fact-checking websites and any sites that disallow automated bot access. A list of excluded domains can be found in Appendix D. For each retrieved URL, we scrape the corresponding page using Firecrawl[6]. Unlike previous work, we extend the scraper to identify and download any referenced image, ensuring a complete context for the fact-check.

**Stage 3: Summarize Results.** At this stage, the gathered evidence is integrated into the fact-checking report, which guides the MLLM's reasoning through in-context learning. The model generates an abstractive summary of key findings for each tool output, ensuring brevity and alignment with the existing report. Relevant images are retrieved and incorporated, while irrelevant results are filtered by instructing the MLLM to return NONE if they do not contribute meaningfully to the verification process.

**Stage 4: Develop the Fact-Check.** Corresponding to Stage 4 in Moreno Gil et al. (2021), DEFAME brings claim and summarized evidence together. It directs the MLLM to discuss the claim's veracity step-by-step based on the evidence, flagging any gaps as "incomplete" if the information is missing. This stage offers room for intricate reasoning, deducing new insights through natural language inference. It serves as an immediate preparation for the following stage.

**Stage 5: Predict a Verdict.** Next, DEFAME classifies the claim into one of the benchmark-specific categories by

---

[4]https://serper.dev

[5]https://cloud.google.com/vision/

[6]https://github.com/mendableai/firecrawl

prompting the MLLM to summarize key findings and select a verdict. If the model returns ❓ **NEI** (Not Enough Information), the system loops back to Stage 1 to retrieve additional evidence, enabling deeper exploration of unresolved aspects. The process proceeds to the final stage once a definitive verdict is reached or three iterations have been completed, reflecting the iterative nature of human fact-checking.

**Stage 6: Justify the Verdict.** Human readability and explainability are essential to effective fact-checking. While the fact-checking report provides a detailed and transparent account of the verification process, its length can become very long—potentially overwhelming users seeking a quick understanding of the outcome. This final, post-prediction stage addresses that need by generating a concise summary that distills the key findings and critical evidence, including hyperlinks. To this end, the MLLM is prompted with the complete report and tasked with producing a focused, readable justification. The resulting summary is appended to the full report, serving both as an accessible explanation for end users and as a support tool for further human verification.

## 4. CLAIMREVIEW2024+

We introduce CLAIMREVIEW2024+, a novel benchmark designed to evaluate fact-checking models on claims that fall outside the pretraining scope of current MLLMs, avoiding the data leakage issue (see Appendix A for more details). To this end, we curate a set of 300 English claims from the Google FactCheck Claim Search API[7], which indexes professionally fact-checked claims published by organizations such as Snopes, PolitiFact, and AFP via the ClaimReview markup standard[8]. All claims are sampled from fact-checking articles from the period between November 1, 2023, and January 18, 2025—notably, after the October 2023 knowledge cutoff of GPT-4O—enabling us to simulate a realistic, temporally out-of-distribution evaluation setting. The dataset consists of 160 unimodal (text-only) and 140 multimodal (text-image) claims (cf. Fig. 3).

Claims are retrieved via keyword-based queries across ten broad topics (e.g., politics, health, climate) and de-duplicated based on review URLs. For each instance, we extract the date and the claimant for a clearer claim context. If the claim date is not present in the markup, we use the review's release date instead. Labels are adapted from those provided by the fact-checking organizations mapped to a unified four-class taxonomy inspired by AVERITEC: ✅ **Supported**, ❌ **Refuted**, ⚡ **Misleading**, and ❓ **NEI**—see Appendix B for the specific class definitions. Label assignment follows a two-stage process: Trivial cases are mapped automatically using an LLM-based script (included in the

---

[7] https://toolbox.google.com/factcheck/apis
[8] https://www.claimreviewproject.com/

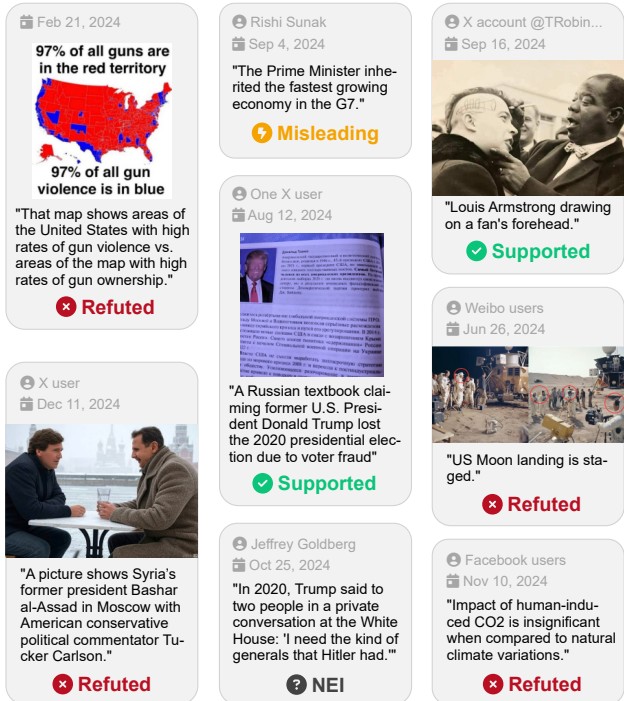

Figure 3: Examples from our CLAIMREVIEW2024+ benchmark, containing claimant, claim date, claim, and verdict.

release), while ambiguous ones are manually labeled by a PhD-level AFC researcher. The dataset is validated by a second annotator, comparing all instances with the original fact-checking articles directly.

To prevent leakage of veracity judgments, we manually edit claims that contain explicit fact-checking language (e.g., rewriting "Fact Check: Image from Pakistan falsely claims..." to "Image from Pakistan shows..."). Since the ClaimReview API often returns teaser images with overlays or composites, all images are manually curated to reflect the original visuals referenced in the claims.

The final label distribution is: 129 ❌ **Refuted**, 89 ✅ **Supported**, 61 ⚡ **Misleading**, and 21 ❓ **NEI**. While we ensure all claims are posted after the GPT-4O cutoff date, we note that some may represent earlier events re-evaluated by fact-checking organizations. Given its source diversity and topical focus on timely, high-impact misinformation, we believe CLAIMREVIEW2024+ offers a challenging, real-world testbed for current fact-checking models.

## 5. Experiments

### 5.1. Datasets

Next to CLAIMREVIEW2024+, we evaluate DEFAME on three well-known fact-checking datasets, representative of distinct areas in fact-checking literature.

| | **DEFAME** backbone | | | |
|---|---|---|---|---|
| **Dataset** | GPT-4O | GPT-4O MINI | LLAVA-1V | LLAMA 4 |
| AVERITEC | **70.5** | 68.8 | 49.3 | 67.0 |
| MOCHEG | **59.2** | 55.5 | 42.1 | 55.0 |
| VERITE | **83.9** | 67.1 | 59.3 | 72.3 |
| CLAIMREVIEW2024+ | **69.7** | 47.7 | 32.6 | 48.8 |

Table 2: DEFAME accuracy with different backbone MLLMs (columns) on the four benchmarks (rows).

**AVERITEC** (Schlichtkrull et al., 2024b) is a popular text-only real-world-based benchmark. The development set consists of 500 claims: 305 ❌**Refuted**, 122 ✅**Supported**, 35 ❓**NEI** (Not Enough Information), and 38 claims with the ⚡**C/CP** (Conflicting/Cherrypicking) label that designates claims with conflicting evidence or claims that are technically true but lack context. We retrieve evidence from the benchmark-complementary Knowledge Base (KB), which contains the necessary evidence along with approximately $1,000$ unrelated resources to simulate open web search. Thus, for AVERITEC, the Web Search Tool does not utilize the Serper API but rather a semantic search, yielding 5 results for each search. Each query to the KB is encoded using `gte-base-en-v1.5` (Alibaba-NLP, 2024); the closest documents to the search query are retrieved via $k$-nearest neighbor. We report the accuracy over all 4 classes.

**MOCHEG** (Yao et al., 2023) features textual claims paired with text-image evidence. Its multimodal nature qualifies it as a benchmark for DEFAME. Out of the $2,001$ unique claims in the test set, we choose the $1,689$ claims that have a final ruling, useful to assess the quality of generated justifications (Appendix M). That subset includes 667 ❌**Refuted**, 522 ❓**NEI**, and 500 ✅**Supported** claims. We evaluate our performance using accuracy—equivalent to micro-F1 (Appendix J).

**VERITE** (Papadopoulos et al., 2024b) is an image-text verification benchmark focused on Out-Of-Context (OOC) scenarios. After removing 13 incomplete instances, VERITE comprises $1,001$ samples, sourced partly from fact-checking platforms and partly generated by swapping images or altering captions. The dataset includes 338 ✅**True**, 325 ❌**OOC**, and 338 ❌**Miscaptioned** claims (OOC and miscaptioned claims differ in construction but both involve out-of-context imagery). Following Papadopoulos et al. (2024a), we report accuracy for "✅**True** vs. ❌**OOC**" and "✅**True** vs. ❌**Miscaptioned**," as well as a merged "✅**True** vs. ❌**False**" setup.

### 5.2. Model and Configuration

We chose GPT-4O and GPT-4O MINI as the backbone of DEFAME since they are the current state-of-the-art

MLLMs. To account for open-source MLLMs, we also test LLAVA-ONEVISION (1V) (7B) (Li et al., 2024a), and LLAMA 4 SCOUT (Meta AI, 2025). DEFAME includes the MLLM without any fine-tuning, with temperature set to $0.01$ and top-$p$ to $0.9$ to control response diversity. We limit the number of images per scraped web page to a maximum of 32 to avoid an excessive flood of images. DEFAME processes interleaved text-image inputs, preserving the original position of images within the text context, but any input exceeding the MLLM's maximum context window is truncated accordingly. Table 2 shows the performance of the different backbones. LLAVA-ONEVISION (7B) struggled with formatting outputs consistently for downstream use, leading to a substantial performance drop. While LLAMA 4 SCOUT addressed this issue and achieved results comparable to GPT-4O MINI—suggesting that open-source models are gradually closing the gap—GPT-4O still outperformed all other backbones by a wide margin, particularly on CLAIM-REVIEW2024+ and VERITE.

### 5.3. Comparison to the State-of-the-Art

In Table 3, we present our evaluation results compared to State-of-the-Art (SOTA) methods and two baselines: GPT-4O, which directly generates a verdict ("Determine the claim's veracity by picking one of the following decision options..."), and GPT-4O Chain-of-Thought (CoT), which also relies solely on parametric knowledge but is instructed to perform step-by-step reasoning before generating the verdict. For robust comparison, we run both baselines and our main variant three times on each dataset and report the mean performance. DEFAME achieves an accuracy of $70.5\%$ on the AVERITEC benchmark and surpasses the previous SOTA (Cao et al., 2023) that deploys a CoT LLM approach. With an overall "✅**True** vs. ❌**False**" accuracy of $83.9\%$ on VERITE, DEFAME ranks 25.9 percentage points above prior best result (Papadopoulos et al., 2024a), similar for "✅**True** vs. ❌**Miscaptioned**". It performs competitively on the "✅**True** vs. ❌**OOC**" accuracy. On MOCHEG, our framework achieves an average accuracy of $59.2\%$, replacing the METASUMPERCEIVER (Chen et al., 2024) as the new SOTA fact-checking system.

Importantly, no prior work has demonstrated the ability to *simultaneously* address such diverse tasks as we do here—making DEFAME the most general AFC method as of now. Unlike prior methods that are limited to specific subtasks or modalities—such as OOC detection, text-only verification, or reliance on gold evidence—DEFAME performs robustly across all benchmarks without task-specific tuning or training data. This benchmark-spanning performance is made possible by DEFAME's zero-shot, retrieval-based design, which does not depend on gold-labeled evidence, modality constraints, or dataset-specific preprocessing.

| Method | AVERITEC | MOCHEG | VERITE | | | CR+ |
|---|---|---|---|---|---|---|
| | Acc | Acc | T/OOC | T/MC | T/F | Acc |
| CFR            (Sriram et al., 2024) | 60.0 | - | - | - | - | - |
| DEBERTA            (Cao et al., 2023) | 65.6 | - | - | - | - | - |
| LVLM4FV     (Tahmasebi et al., 2024) | - | 45.1 | - | - | - | - |
| METASUM            (Chen et al., 2024) | - | 48.6 | - | - | - | - |
| CHASMA (Papadopoulos et al., 2024b) | - | - | 74.4* | 59.3* | 52.1 | - |
| AITR       (Papadopoulos et al., 2024a) | - | - | **82.7** | 51.8 | 58.0* | - |
| GPT-4O | 62.1 ±0.4 | 53.7 ±0.4 | 70.4 ±1.4 | 72.0 ±0.8 | 78.7 ±0.8 | 35.2 ±0.9 |
| GPT-4O COT | 61.0 ±0.4 | 49.7 ±0.3 | 74.1 ±0.9 | 76.5 ±0.4 | 80.0 ±0.6 | 31.4 ±4.5 |
| **DEFAME (Ours)** | **70.5** ±0.6 | **59.2** ±0.4 | 78.4 ±1.0 | **83.3** ±1.1 | **83.9** ±0.5 | **69.7** ±2.5 |

Table 3: Comparison of our method with prior state-of-the-art methods and baselines across datasets. Best scores are in **bold**, second-best are underlined. For DEFAME and the GPT-4O baselines, we report the mean over three runs along with the standard deviation. Values marked with * were derived from reported numbers. Blank cells indicate that the respective method could not be evaluated on the corresponding benchmark—either due to unavailability of publicly released code or incompatibility with the benchmark setup. CR+ = CLAIMREVIEW2024+.

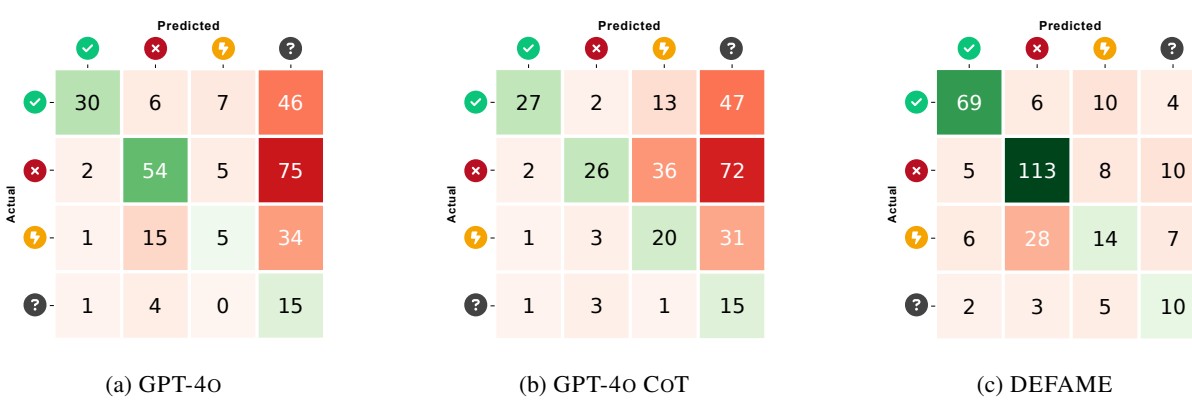

(a) GPT-4O

(b) GPT-4O COT

(c) DEFAME

Figure 4: Confusion matrices on the CLAIMREVIEW2024+ dataset for GPT-4O, GPT-4O CoT, and DEFAME.

The results on our new CLAIMREVIEW2024+ dataset challenge the notion that vanilla LLMs are reliable fact-checking systems. While performance on established benchmarks is strong, CLAIMREVIEW2024+ reveals a drastic drop for the two GPT-4O baselines, whereas DEFAME maintains accuracy comparable to its performance on AVERITEC. This suggests that DEFAME's evidence retrieval mitigates the temporal dependence of its backbone model. The confusion matrices (Figure 4) illustrate distinct error patterns across models on the CLAIMREVIEW2024+ dataset. Pure GPT-4O overpredicts ❓NEI but outperforms its Chain-of-Thought counterpart in identifying ❌Refuted claims. GPT-4O CoT also acknowledges the lack of evidence but shows a stronger tendency to commit to ⚡Misleading. In contrast, DEFAME exhibits a more balanced confusion matrix, with its primary challenge lying in differentiating ⚡Misleading and ❌Refuted—a relatively uncritical ambiguity among the possible misclassifications. Its result on the ✅Supported class is far superior to the two baselines. Refer to Appendix G for further confusion analysis.

Figure 5 exemplarily shows the claim "Slovakian Prime Minister Robert Fico being dragged into a car after being shot." The GPT-4O COT baseline refutes it due to missing evidence. In contrast, DEFAME retrieves supporting articles from CNN, Vatican News, and Al Jazeera via reverse image and web search, confirming that Fico was indeed shot on May 15, 2024, in Handlová, Slovakia, and subsequently moved into a car. Although the geolocation suggested Poland or the Czech Republic, DEFAME correctly marginalized this noise and predicted the correct verdict.

### 5.4. Ablation Study

We conduct an ablation study with reduced DEFAME variants to investigate the contributions of various components and capabilities of DEFAME to its overall performance:

- **Tool Ablations:** We assess the contribution of each of the four tools by individually removing them from the tool pool.

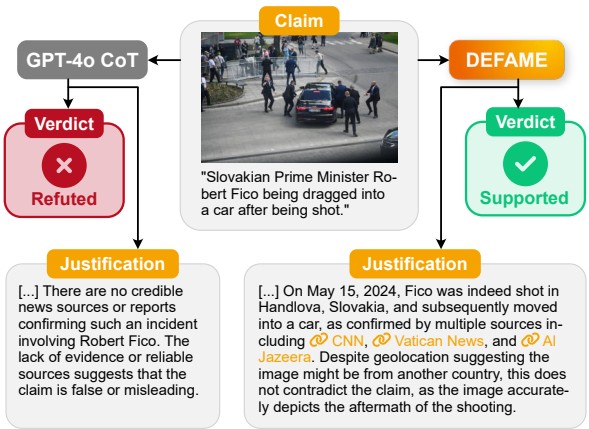

Figure 5: Example claim from CLAIMREVIEW2024+ with verdict and justification by GPT-4O CoT and DEFAME.

| Model Variant | MOCHEG Acc. | VERITE T/F (Acc.) | CR+ Acc. |
|---|---|---|---|
| DEFAME | **59.2** | **83.9** | **69.7** |
| w/o Geolocation | 58.3 | 80.6 | 65.7 |
| w/o Reverse Search | 58.2 | 73.7 | 64.0 |
| w/o Image Search | 57.8 | 81.4 | 63.7 |
| w/o Web Search | 42.0 | 81.8 | 59.7 |
| Single Turn | 47.7 | 82.8 | 63.3 |
| w/o Planning | 58.7 | 83.0 | 68.0 |
| w/o Develop | 57.4 | 83.8 | 67.0 |
| Unimodal Develop | 56.1 | 82.0 | 65.7 |

Table 4: Ablation study results for different model variants on the MOCHEG, VERITE and CLAIMREVIEW2024+ datasets. "✅True vs. ❌False" accuracy is reported for VERITE. Best scores are marked in **bold**, second best are underlined.

- **Single Turn:** This version is restricted to pass all stages only once, i.e., no ability to delve deeper into findings.

- **W/o Planning:** This variant fixes all actions into a static action schedule. Each tool is executed exactly once, bypassing the dynamic planning stage.

- **W/o Develop Stage:** Is intermediate reasoning useful? This variant excludes the Develop Stage, jumping from evidence retrieval immediately to judgment.

- **Unimodal Develop Stage:** Is textual reasoning sufficient? This part-unimodal variant can retrieve evidence images but cannot pass them on to future stages.

Table 4 presents the results of our ablation study. We observe that DEFAME benefits from all four tools—with varying contributions: Web Search is critical for MOCHEG and

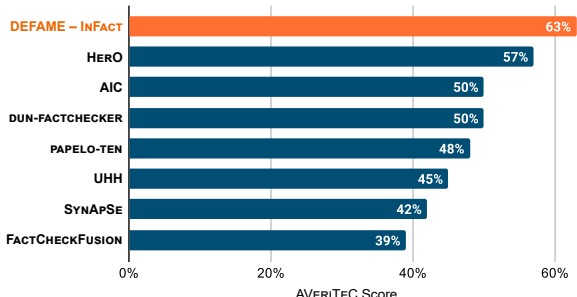

Figure 6: Top 8 (of 21) systems on the AVERITEC Challenge test set, ranked by AVERITEC score.

CLAIMREVIEW2024+, likely because both contain only or many text-only claims, respectively. Geolocation and Reverse Image Search prove more useful for VERITE, where images are central and can be directly examined.

The single-turn variant of DEFAME performs notably worse than the multi-turn version, especially on MOCHEG, underscoring the importance of follow-up retrievals. The Develop Stage and the integration of visual evidence from tool outputs into the report yield a clear performance gain. Finally, removing the Planning Stage leads not only to reduced accuracy but also to significantly higher computational costs (see Appendix C).

### 5.5. Adaptation to the AVERITEC Challenge

The AVERITEC authors recently hosted a shared task (Schlichtkrull et al., 2024a), where systems competed based on the AVERITEC Score (Schlichtkrull et al., 2024b), a metric that evaluates both claim accuracy *and* evidence correctness. To participate, we adapted our framework to produce question-answer (QA) pairs, as the metric relies on the similarity between generated and reference QA pairs. Our adapted method, submitted under the name **INFACT**, achieved best performance in the AVERITEC Challenge, as shown in Figure 6 (see also Appendix E and Rothermel et al. (2024) for details). Notably, both the original DE-FAME and the adapted INFACT variant achieve comparable accuracy on the development set, demonstrating that our system is not only robust across evaluation protocols but also flexible enough to support output formats required by the downstream tasks.

### 5.6. Explainability Quality and Human Evaluation

To further measure the quality of gathered evidence and to assess our framework's vulnerability to hallucinations, we conducted a human evaluation, focusing two aspects:

- **Coherence:** The fact-check maintains a logical and meaningful flow. There are no contradictions or gaps

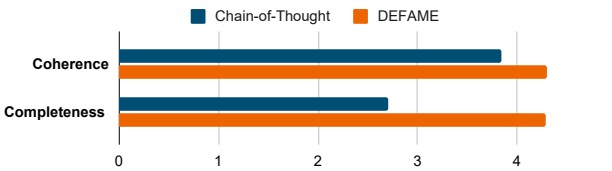

Figure 7: Human assessment of Coherence and Completeness of DEFAME's vs. GPT-4O COT's fact-checking reports on 36 claims sampled from three benchmarks.

that disrupt the overall coherence of the report.

- **Completeness:** The verdict is sufficiently justified: he included evidence allows one to derive the verdict.

Participants were asked to rate the fact-checking reports w.r.t. the above criteria on a Likert scale from 1 to 5. In total, we collect 185 ratings over 36 claims sampled randomly but balanced from VERITE, MOCHEG, and AVERITEC. Please refer to Appendix L for further details on the experiment. The results are shown in Fig. 7. The CoT baseline and DEFAME show no significant difference in output coherence, i.e., both produce mostly logical reports. This is expected, considering the current state of zero-shot LLMs in text generation. However, the two differ significantly in their ability to justify their verdicts. The results imply that DEFAME provides better justifications compared to bare MLLM prompting. This finding further challenges recent studies (Geng et al., 2024; Beigi et al., 2024; Shao et al., 2023; Li et al., 2024e) that suggest MLLMs can perform fact-checking without retrieving external evidence.

### 5.7. Failure Analysis

We analyzed 119 VERITE and CLAIMREVIEW2024+ instances that were mislabeled by DEFAME and identified five common failure modes. **(1) Label Ambiguity**, the most frequent one, occurs when DEFAME mixes up similar labels like ⊗**Refuted** and ⚡**Misleading**. **(2) Missed Evidence** accounts for errors where DEFAME exhausts all three retrieval attempts but fails to find enough evidence. This happens often when evidence is inaccessible (e.g., when proprietary API access is required) or embedded in a video. **(3) Reasoning Errors** are mistakes that happen during natural language inference, e.g., mixing up numbers. **(4) Premature Judgment** occurs when DEFAME commits to a verdict before gathering sufficient evidence. For example, it predicts ✅**Supported** as soon as it finds *any* image matching the claim text, even if it is not *the one* in the claim. **(5) Wrong Ground Truth**: In several cases, our inspection revealed annotation errors, meaning DEFAME's prediction was actually correct. Appendix I shows instances with wrong ground truth labels.

## 6. Discussion

While DEFAME works with any MLLM backbone, the backbone's capabilities directly influence the quality of the resulting fact-check. As MLLMs continue to evolve, DEFAME's performance is expected to improve correspondingly. Moreover, its modular, in-context-learning based design enables DEFAME to incorporate further tools beyond those demonstrated in this work. Still, DEFAME has several limitations.

**Credibility of External Evidence:** Our reliance on search engines introduces the risk of incorporating unreliable information, as search results may include leaked or biased content (Chrysidis et al., 2024). While some approaches assess source credibility using third-party ratings (Zhou et al., 2024), our framework relies on diversification and the search engine's internal trustworthiness checks[9]. Introducing external ratings could strengthen verification rigor.

**System Stability and Sensitivity:** Our web scraping process is prone to instability due to restricted access and large document size. Moreover, open-source models like LLAVA exhibit formatting sensitivity, where minor prompt variations impact response quality. Ensuring robust scraping and prompt formatting stability is crucial for reliable outputs.

**Hallucination:** We acknowledge that hallucinations are a risk in DEFAME as they are inherent to LLMs. However, both qualitative analysis and human evaluation do not indicate substantial amounts of hallucinations. Still, we agree that the role of hallucination in DEFAME must be analyzed more closely in future work.

## 7. Conclusion

We presented DEFAME, a comprehensive zero-shot framework for multimodal fact-checking that integrates MLLMs with external tools to address the limitations of traditional automated fact-checking approaches. Our framework explainably grounds its analysis in verifiable data by combining MLLM-driven reasoning with external multimodal evidence sources, geolocation, reverse image search, and dynamic web searches. We demonstrated that DEFAME achieves new state-of-the-art results across multiple diverse benchmarks, including an own leakage-free benchmark, highlighting its generality and capability to navigate unseen real-world claims. While limitations remain, our system represents a significant advancement in the field, providing a versatile foundation for future developments in automated multimodal fact-checking.

---

[9] https://blog.google/products/search/how-google-delivers-reliable-information-search/

## Impact Statement

To scalably debunk the masses of mis- and disinformation—which indisputably pose a substantial threat to social cohesion—automating fact-checking is inevitable. Although the results do not sufficiently warrant real-world application yet, DEFAME's introduction is a step in addressing that challenge.

DEFAME contributes to shifting the paradigm toward explainable and verifiable AI systems. Its ability to provide structured, evidence-backed justifications ensures that AFC does not become a black-box authority but rather an augmentation tool that strengthens public resilience against misinformation. By embedding transparency and iterative verification into its core process, DEFAME helps counter the skepticism toward AI-driven moderation while simultaneously reducing the burden on human fact-checkers.

However, the widespread deployment of such systems also raises ethical and societal concerns. Recently, in public discussions, professional fact-checkers have widely faced accusations of being biased (CNN, 2025) and, therefore, unreliable. Automated fact-checking, even when well-calibrated, can also shape narratives by prioritizing certain sources or by amplifying institutional biases inherent in retrieval mechanisms or parametric knowledge. Current research efforts strive to de-bias LLMs (Gallegos et al., 2024), which DEFAME will automatically benefit from thanks to its backbone-agnostic design. Within the strongly polarized public discussions, AFC systems may emerge as a nonpartisan information source. Their reliability inherently depends on the quality of the backbone, which is difficult to rigorously analyze.

There is also a risk that governments or platforms might misuse systems like DEFAME for overreach in content moderation, potentially stifling dissent or critical journalism under the guise of misinformation control. To mitigate this, it is crucial to ensure accountability, source diversity, and contestability—allowing users to challenge and scrutinize automated decisions. Additionally, while DEFAME may improve trust in digital information, its success will ultimately depend on how it is integrated into broader fact-checking ecosystems and content moderation, where human oversight remains essential. The current debate strikingly shows that fact-checking (especially of social media content) sits on the verge between public safety and perceived censorship of free speech. Therefore, the real impact of DEFAME lies not just in its technical contributions but in how humans will ultimately use it.

## Acknowledgments

We thank **Marcus Kornmann** for his diligent assistance in conducting this research.

The research was partially funded by a **LOEWE-Start-Professur** (LOEWE/4b//519/05.01.002-(0006)/94), **LOEWE-Spitzen-Professur** (LOEWE/4a//519/05.00.002-(0010)/93) and an **Alexander von Humboldt Professorship in Multimodal Reliable AI**, sponsored by Germany's Federal Ministry for Education and Research.

For compute, we gratefully acknowledge support from the **hessian.AI Service Center** (funded by the Federal Ministry of Education and Research, BMBF, grant no. 01IS22091) and the **hessian.AI Innovation Lab** (funded by the Hessian Ministry for Digital Strategy and Innovation, grant no. S-DIW04/0013/003).

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

# Appendix

The appendix provides an in-depth extension of the main paper, showcasing additional examples, analyses, and details that complement the findings discussed. The data leakage problem of existing AFC benchmarks is discussed in more detail in Section A. In Section B, the definitions of the four CLAIMREVIEW2024+ classes are presented. We include information on compute and cost in Section C. Section D outlines excluded domains to ensure a fair and realistic evaluation setting. Details about our adaptations for the AVERITEC challenge are given in Section E. To enhance clarity, we formalize the iterative structure and processing stages of DEFAME in Appendix F. A short analysis of the confusions committed by the CoT variant and DEFAME is shown in Section G. Section H includes the prompts used in DEFAME. Section I identifies wrongly annotated samples in the VERITE dataset. Section J provides a high-level derivation showing the equivalence of micro-F1 and accuracy in single-label multiclass settings, clarifying a common source of confusion in prior evaluations. Section K discusses cases with outdated ground truth in the MOCHEG dataset and their implications. Section L specifies our human evaluation of the fact-checking report. In Section M, we analyze the justifications generated by DEFAME and scrutinize current evaluation metrics. Section N illustrates example fact-checking reports produced by DEFAME. Lastly, Section O delves into failure cases, highlighting common pitfalls and their underlying causes.

## A. The Data Leakage Issue

Most AFC benchmarks, particularly AVERITEC, VERITE, and MOCHEG, are (largely) composed of real-world claims sourced from actual fact-checking articles on the web. The covered time spans vary:

- AVERITEC claims stem from the time until December 2021.

- VERITE covers claims between January 2001 and January 2023.

- MOCHEG contains claims up until May 2022.

Importantly, all time ranges lie before GPT-4O's knowledge cutoff date, which is October 2023. Since the claims are publicly fact-checked, it is highly likely that GPT-4O was pretrained on large portions of the claims *and verdicts* contained in these benchmarks, a matter of **data leakage**. Therefore, we introduced CLAIMREVIEW2024+, which consists of claims, the fact-checks of which were published only after the knowledge cutoff date.

## B. CLAIMREVIEW2024+ Class Definitions

CLAIMREVIEW2024+ uses four classes, defined as follows:

- ⊗ **Refuted**: "A claim is considered refuted when the evidence contradicts the claim."

- ✓ **Supported**: "The claim is accurate based on evidence."

- ⚡ **Misleading**: "The claim is misleading or requires additional context."

- ❓ **NEI**: "The claim does not have enough information to be verified."

For all datasets, the class definitions were provided to DE-FAME at the judgment stage.

## C. Details on Compute and Cost

As the GPT models are available only via OpenAI's API, most of our computation happens externally. On our end, we employed four NVIDIA A100-80GB GPUs in order to execute LLAVA-1V and GEOCLIP. All other processing was performed on 32 AMD EPYC 7313 16-core CPUs. Since fact-checks on VERITE claims are multimodal from the start, we chose VERITE as a representative surrogate for the setting of multimodal AFC. We thus report usage statistics on VERITE in Table 5.

As expected, the naive baselines (Pure GPT-4O and CoT) require the least amount of resources. But, as discussed in Section 5, this is due to the absence of any external evidence and comes at the cost of lower accuracy. DEFAME with GPT-4O MINI and LLAVA-1V as the backbone MLLM runs faster *and* cheaper than DEFAME with GPT-4O, but yields decreased accuracy as well. Surprisingly, LLAVA-1V fact-checks faster than the GPT models. We found the reason to lie in LLAVA-1V's inability to correctly format the proposed actions, yielding a smaller number of executed actions and, thus, shorter fact-checks overall. LLAMA 4 SCOUT masters action formatting and is cheaper than the GPT-4O models but—surprisingly—takes drastically more time. We have not taken any acceleration measures beyond basic LLM installation, which may leave room for time improvement.

Compared to human fact-checking experts—who invest about an entire working day to debunk a claim and write a corresponding article (Hassan et al., 2015)—DEFAME's fee of $0.13 per claim is cheap. However, DEFAME's output quality does not match human fact-checkers yet. Thus, DEFAME could serve as a cost-effective assistant to aid human fact-checkers. Theoretically, social media platforms could also use DEFAME to cheaply mass-check larger

| Method | | Time min / Claim | LLM API Cost ¢ / Claim | Search API Cost ¢ / Claim | # Search Queries Searches / Claim | # Input Tokens Processed Tokens / Claim | # Output Tokens Generated Tokens / Claim |
|---|---|---|---|---|---|---|---|
| **DEFAME** | GPT-4O | 4:14 | 13 | <1 | 2.2 | 48K | 921 |
| | GPT-4O MINI | 3:13 | <1 | <1 | 3.1 | 54K | 1300 |
| | LLAVA-1V | 1:23 | - | <1 | 0.9 | 14K | 794 |
| | LLAMA 4 | 18:02 | - | <1 | 2.8 | 21K | 1325 |
| **DEFAME Ablations** | w/o Geolocation | 3:12 | 11 | <1 | 2.3 | 40K | 910 |
| | w/o Reverse Search | 2:37 | 8 | <1 | 2.3 | 28K | 828 |
| | w/o Image Search | 2:57 | 11 | <1 | 2.2 | 40K | 909 |
| | w/o Web Search | 2:30 | 10 | <1 | 2.2 | 37K | 761 |
| | Single Turn | 2:47 | 10 | <1 | 2.0 | 38K | 858 |
| | w/o Planning | 5:11 | 14 | <1 | 5.2 | 70K | 1100 |
| | w/o Develop | 1:56 | 10 | <1 | 2.1 | 42K | 710 |
| | Unimodal Develop | 3:15 | 10 | <1 | 2.3 | 41K | 865 |
| **Baselines** | GPT-4O | 0:06 | <1 | 0 | 0.0 | 2.2K | 67 |
| | GPT-4O COT | 0:09 | <1 | 0 | 0.0 | 2.3K | 177 |

Table 5: Usage and computational cost statistics of DEFAME and all considered variants (ablations and baselines) on VERITE. Time values are rough estimates and are subject to high variance due to hardware usage by other processes.

amounts of claims posted online. However, depending on the number of claims—which, on social media, arguably exceed millions per day—DEFAME could become expensive. Claim filtering approaches would be needed to narrow down the claims to the most check-worthy ones in order scale DEFAME up to larger amounts.

## D. Excluded Domains

To maintain a realistic and fair setting, we exclude all major fact-checking organizations we know from all web search results. Table 6 shows the corresponding list of domains. Additionally, several platforms forbid (direct) automatic access to their web pages, cf. Table 7. Any URL with a sub-string matching the domains and URLs in the Tables mentioned earlier is removed from the search results and, hence, ignored by the fact-check.

## E. Adaptation to the AVERITEC Challenge

The AVERITEC challenge evaluates fact-checking quality using the AVERITEC Score that compares model-generated question-answer (QA) pairs with gold QA pairs provided in the benchmark (Schlichtkrull et al., 2024b). To perform well, a method must effectively identify and address *the same (or very similar) questions and answers posed by the benchmark annotators*.

To align with this evaluation, we introduce an extension called INFACT. This adaptation begins by generating 10 key

| **Excluded Fact-Checking URLs** |
|---|
| snopes.com |
| politifact.com |
| factcheck.org |
| truthorfiction.com |
| fullfact.org |
| leadstories.com |
| hoax-slayer.net |
| checkyourfact.com |
| reuters.com/fact-check |
| reuters.com/article/fact-check |
| apnews.com/APFactCheck |
| factcheck.afp.com |
| poynter.org |
| factcheck.ge |
| vishvasnews.com |
| boomlive.in |
| altnews.in |
| thequint.com/news/webqoof |
| factcheck.kz |

Table 6: List of excluded URLs to maintain a fair and realistic fact-checking setting.

| Unsupported Domains |
| --- |
| facebook.com |
| twitter.com |
| x.com |
| instagram.com |
| youtube.com |
| tiktok.com |
| reddit.com |
| ebay.com |
| microsoft.com |
| researchhub.com |
| pinterest.com |
| irs.gov |

Table 7: List of excluded domains due to bot traffic restrictions.

questions designed to probe the claim's veracity. The Planner then proposes targeted search queries for each question and applies the Web Search tool to retrieve up to 5 relevant search results. Using the retrieved evidence, the LLM backbone attempts to answer the questions systematically. Finally, the system synthesizes the answers into a coherent verdict, ensuring the reasoning is grounded in the collected evidence. The resulting QA pairs are evaluated using the AVERITEC Score, showcasing INFACT 's alignment with the benchmark's evaluation criteria while maintaining its structured, evidence-driven methodology.

## F. Formal Representation of DEFAME

We include a formalization of the DEFAME pipeline to clarify the role of each stage and its iterative structure.

Let $\mathcal{T}$ and $\mathcal{I}$ denote the spaces of text and images, respectively. Define $\mathcal{M} := (\mathcal{T} \cup \mathcal{I})^*$ as the space of multimodal sequences, and let $\mathcal{Y}$ denote the space of verdict labels. Then, DEFAME is a function

$$\mathcal{F} : \mathcal{M} \to \mathcal{M} \times \mathcal{Y}, \quad \mathcal{F}(c) = (R_{\text{out}}, y_{\text{out}}),$$

where, given a claim $c \in \mathcal{M}$, the output consists of a report $R_{\text{out}}$ containing the full fact-check and a predicted verdict $y_{\text{out}}$. DEFAME proceeds iteratively up to $N$ steps. We can denote each iteration with $\mathcal{F}_{\text{iter}} : \mathcal{M} \to \mathcal{M} \times \mathcal{Y}$, so that

$$(R^{(i+1)}, y^{(i+1)}) := \mathcal{F}_{\text{iter}}(R^{(i)}),$$

i.e., an (incomplete) report $R^{(i)} \in \mathcal{M}$ gets extended with new actions, evidence, and elaboration, resulting in report $R^{(i+1)}$ and intermediate verdict $y^{(i+1)}$. We can decompose $\mathcal{F}_{\text{iter}}$ into the five individual pipeline stages

$$\mathcal{F}_{\text{iter}} := \mathcal{S}_5 \circ \mathcal{S}_4 \circ \mathcal{S}_3 \circ \mathcal{S}_2 \circ \mathcal{S}_1,$$

where each stage can be described as follows:

1. **Planning ($\mathcal{S}_1$):** Select actions $A \subseteq \mathcal{A}$ based on the current report $R^{(i)}$.

2. **Execution ($\mathcal{S}_2$):** Retrieve evidence $E := \{\tau(a) \mid a \in A\}$, where $\tau$ executes the tool action $a$.

3. **Summarization ($\mathcal{S}_3$):** $R_1^{(i)} := \sigma(E, R^{(i)})$, where $\sigma$ summarizes $E$ in context and appends it to the report.

4. **Develop ($\mathcal{S}_4$):** $R_2^{(i)} := \mathcal{S}_4(R_1^{(i)})$, where $\mathcal{S}_4$ generates structured reasoning and expands the report.

5. **Verdict Prediction ($\mathcal{S}_5$):** $(R_3^{(i)}, y^{(i)}) := \mathcal{S}_5(R_2^{(i)})$, returning an updated report $R_3^{(i)} \in \mathcal{M}$ and a verdict $y^{(i)} \in \mathcal{Y}$.

Let $i^* := \min\{i \leq N \mid y^{(i)} \neq \text{NEI or } i = N\}$, then the final outputs are $y_{\text{out}} := y^{(i^*)}$ and $R_{\text{out}} := \mathcal{S}_6(R^{(i^*)})$, where $\mathcal{S}_6$ denotes the **justification** stage that appends a rationale to the final report.

This formal view captures the iterative and modular nature of DEFAME, highlighting how evidence is retrieved, processed, and transformed into a final verdict and explanation.

## G. DEFAME Confusions

Figure 8 shows the confusion matrices for all three backbones on the four benchmarks, respectively.

A closer look at Figures 8g, 8h and 8i reveals that GPT-4O with and without Chain-of-Thought overpredicts ⊗ **OOC** while DEFAME's confusions are more balanced. Surprisingly, incorporating Chain-of-Thought prompting hurts the performance on MOCHEG (see Figures 8d, 8e and Table 3). According to the confusion matrices the introduction of Chain-of-Thought makes the model more unsure, leaning towards ❓ **NEI** compared to the *Pure* GPT-4O. In contrast, DEFAME predicts too confidently even when the ground truth implies insufficient information. A qualitative analysis of the failure cases reveals that, in several cases, the ground truth explanation is no longer up-to-date (see Sections 5.7 and Appendix I).

On AVERITEC (Figures 8a, 8b, and 8c), we observe the opposite behavior with the GPT-4O variants overpredicting ❓ **NEI** compared to our framework. Lastly, DEFAME's false predictions on the CLAIMREVIEW2024+ dataset appear balanced in Figure 4 while the baselines lack the necessary evidence to make correct veracity predictions, very often defaulting to ❓ **NEI**.

## H. Prompts Used in DEFAME

Each stage of the DEFAME framework is guided by a tailored prompt. These prompts are constructed from prompt

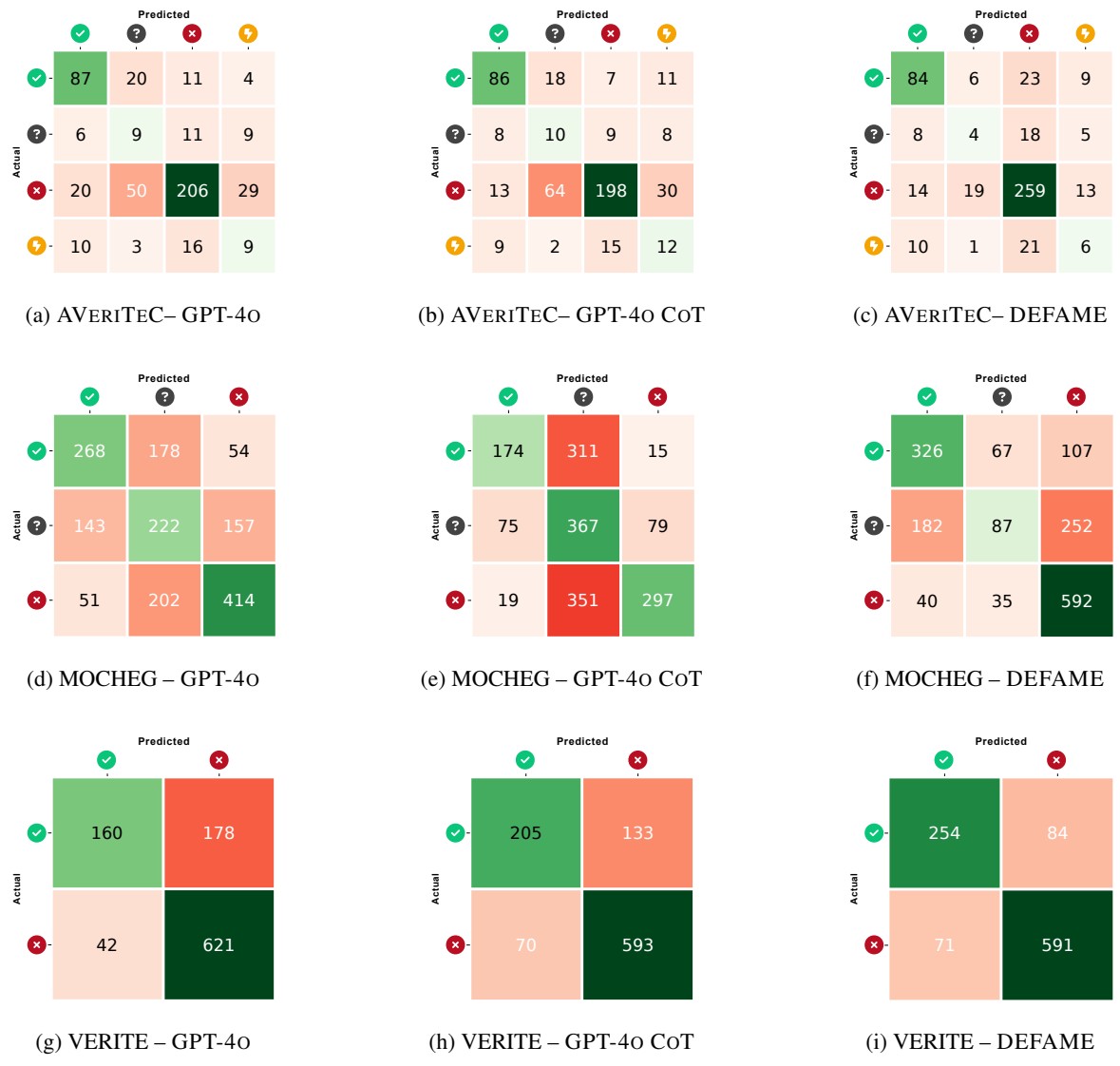

Figure 8: Confusion matrices for GPT-4O, GPT-4O COT, and DEFAME across the AVERITEC, MOCHEG, and VERITE datasets.

templates, where values enclosed in `[]` serve as placeholders. During execution, these placeholders are dynamically replaced with the corresponding variables. This process ensures that the prompt is specific to the current task and context, as illustrated in H.1 and H.2.

In subsequent sections, we present the templates for the remaining four stages of the DEFAME framework. Each template includes detailed explanations of its placeholders. A key placeholder is `[Record]`, which is present in every prompt. This generic placeholder provides the current state of the fact-checking report, consolidating the claim, evidence, and findings gathered so far, ensuring that the LLM operates within the relevant context.

Some prompts require specific values to be returned by the LLM, such as the verdict in the Judge Prompt or the proposed actions in the Plan Prompt. In these cases, both the expected value and its format are explicitly defined within the prompt to guide the LLM's response. Value-specific fallback mechanisms are employed to enhance robustness. These mechanisms, primarily based on regular expressions tailored to observed failure modes, ensure that the required values can still be reliably extracted, even if the LLM deviates from the expected format or introduces minor inconsistencies.

For multimodal LLM inputs and outputs, images are integrated into prompts through a referencing system. When an image reference (e.g., `image:k`) is encountered, the

system inserts a corresponding image block, including the Base64-encoded representation of the image. The LLM references these images using the same identifier, maintaining a consistent link between visual and textual elements.

The prompt templates in DEFAME are dataset-agnostic, enabling use across benchmarks with minimal adaptations. Dataset-specific changes are limited to `[Extra Rules]` in the Plan and Judge Prompts. MOCHEG requires no additional rules, while AVERITEC includes a guideline to avoid the "argument from ignorance" fallacy, ensuring unsupported claims are labeled as `Not Enough Information`. In VERITE, detailed instructions are needed for the ⊗ **OOC** class, which includes samples generated in two ways: true images with altered captions (formerly ⊗ **Miscaptioned**) and true captions with unrelated images. These rules address dataset-specific nuances while maintaining a consistent framework.

## H.1. Plan Prompt Template

> **Instructions**
> The available knowledge is insufficient to assess the Claim. Therefore, **propose a set of actions** to retrieve new and helpful evidence. Adhere to the following rules:
>
> - The actions available are listed under **Valid Actions**, including a short description for each action. No other actions are possible at this moment.
>
> - For each action, use the formatting as specified in **Valid Actions**.
>
> - Include all actions in a single Markdown code block at the end of your answer.
>
> - Propose as few actions as possible but as many as needed. Do not propose similar or previously used actions.
>
> **[Extra Rules]**
> **[Valid Actions]**
> **[Examples]**
> **[Record]**
> **Your Actions:**

- `[Extra Rules]`: Contains benchmark-specific planning guidelines that the Planner must follow when selecting actions. These rules are tailored to the requirements of individual datasets or evaluation scenarios, ensuring that the Planner adheres to task-specific constraints or priorities.

- `[Valid Actions]`: Represents the set of actions available to the Planner at a given stage. The list of valid actions is dynamically adapted to avoid reusing the same action unnecessarily.

- `[Examples]`: Provides in-context examples that demonstrate how to use actions with the correct format. These examples illustrate the structure and logic behind action proposals to guide the Planner.

## H.2. Finalized Plan Prompt

**Instructions**
The available knowledge is insufficient to assess the Claim. Therefore, **propose a set of actions** to retrieve new and helpful evidence. Adhere to the following rules:

- The actions available are listed under **Valid Actions**, including a short description for each action. No other actions are possible at this moment.

- For each action, use the formatting as specified in **Valid Actions**.

- Include all actions in a single Markdown code block at the end of your answer.

- Propose as few actions as possible but as much as needed. Do not propose similar or previously used actions.

- **Consider Both Modalities Equally**: Avoid focusing too much on one modality at the expense of the other, but always check whether the text claim is true or false.

- **Compare Image and Caption**: Verify the context of the image and caption.

**Valid Actions:**

- `geolocate`: Determine the country where an image was taken by providing an image ID.

- `reverse_search`: Perform a reverse image search on the web for similar images.

- `web_search`: Run an open web search for related webpages.

- `image_search`: Retrieve related images for a given query.

**Examples:**

- `geolocate(<image:k>)`

- `reverse_search(<image:k>)`

- `web_search("New Zealand Food Bill 2020")`

- `image_search("China officials white suits carry people")`

**Record:**

Claim: "<image:1232> Image of a bus powered by compressed natural gas, bursting into flames in Italy."

**Your Actions:**

## H.3. Summarize Prompt Template

**Instructions**
In order to find evidence that helps your fact-check, you just ran a web search, which yielded a **Search Result**. **Your task right now is to summarize the Search Result.** What to include:

- Information that might be useful for the fact-check (see **Record**).

- Relevant images (refer to images by inserting their reference in the format `<image:k>`).

- If available: the release date as well as the author or the publisher (e.g., the media company) of the search result.

Do **NOT** include:

- Advertisements.

- Any other information unrelated to the **Record** or the Claim.

**Additional Rules:**

- Do not add any additional information besides the information in the **Search Result**.

- If the **Search Result** doesn't contain any relevant information for the fact-checking work, simply print one word in capital letters: NONE.

- Keep your writing style consistent with the provided Examples.

- Try to filter out relevant information even if the **Search Result** is in a different language.

**[Examples]**

**[Record]**

**[Search_Result]**

**Your Summary:**

- `[Examples]`: Provides 3 in-context examples that demonstrate how to write concise summaries, incorporating relevant images, key insights, and links to sources using Markdown notation. One of the examples shows a case where the search result is irrelevant, guiding the model to return NONE instead of a summary when no useful information is found.

- `[Search_Result]`: Refers to the search result retrieved by a Search tool. This includes the text content scraped from the webpage using the Firecrawl web scraper, the title of the page, any hyperlinks found within the content, images included on the page, and the source URL.

## H.4. Develop Prompt Template

**Instructions**
You just retrieved new Evidence. Now, **analyze the Claim's veracity using the evidence**. Always adhere to the following rules:

- Focus on developing new insights. Do not repeat larger parts from the **Record**. Do not restate the Claim.

- Write down your thoughts step-by-step. Whenever necessary, you may elaborate in more detail.

- Depending on the topic's complexity, invest one to three paragraphs. The fewer, the better.

- If you find that there is insufficient information to verify the Claim, explicitly state what information is missing.

- If you cite web sources, always refer to them by including their URL as a Markdown hyperlink.

- **Use information only from the recorded evidence**: Avoid inserting information that is not implied by the evidence. You may use commonsense knowledge, though.

**[Record]**
**Your Analysis:**

## H.5. Judge Prompt Template

**Instructions**
**Determine the Claim's veracity** by following these steps:

1. Briefly summarize the key insights from the fact-check (see **Record**) in at most one paragraph.

2. Write one paragraph about which one of the **Decision Options** applies best. Include the most appropriate decision option at the end and enclose it in backticks like `this`.

**[Extra Rules]**
**[Decision Options]**
**[Record]**
**Your Judgement:**

- `[Extra Rules]`: Contains benchmark-specific rules or additional constraints that guide the judgment process. These rules are designed to increase the model's understanding of the different classes present in the corresponding benchmark, ensuring consistent verdicts.

- `[Decision Options]`: Lists the possible labels or verdicts that can be assigned to the claim, along with

a short description of each label. These descriptions provide additional context to help the model accurately differentiate between the available options.

## H.6. Justify Prompt Template

**Instructions**
You are provided with the record of a fact-check. It contains the Claim to be verified and documentation of all the fact-checking work along with the gathered evidence. Your task is to **summarize the fact-check**. That is, you provide a concise, one-paragraph justification for the final **VERDICT** based on the knowledge from the **Record**. Note:

- Be truthful, brief, and do not add any additional information besides the information given in the **Record**.

- Link key sources in your summary. Use Markdown notation for that. You may link them in-line.

- Don't state the Claim again. Rather focus on the key insights of the fact-check.

- Simply print just the summary.

**[Record]**
**Summary:**

# I. Wrongly Annotated VERITE Samples

During the qualitative analysis of 20 mispredicted VERITE instances, we encountered 3 cases we argue to be wrongly annotated. Figure 9 shows the corresponding claims and (wrong) annotations. The VERITE annotation classifies the first claim (753) as ✅ **Supported**. However, DEFAME found the image to be captured not in September but in July 2018, citing a fact-check by USA Today[10] from May 2022. Manually investigating this case further, we find that USA Today refers to an article on Mediaite[11] from Sept 6th, 2018, indeed stating that "in July, Clanton shared a photo on Instagram of herself, Thomas, and Ginni Thomas having a 'great weekend' together," showing the screenshot of the respective Instagram post. Hence, the actual correct label for this claim should be ❌ **OOC**.

The wrong annotation can be seen even more clearly for the claims 249 and 250. DEFAME's reverse search yielded multiple credible sources, including an article by CBS News[12], consistently reporting about a gathering of 75 peo-

---

[10] https://web.archive.org/web/20220526225427/https://eu.usatoday.com/story/news/factcheck/2022/05/26/fact-check-photo-ginni-thomas-expensive-wine-2018/9910097002/

[11] https://web.archive.org/web/20180906222511/https://www.mediaite.com/online/exclusive-clarence-thomas-wife-hired-ex-tpusa-staffer-known-for-saying-i-hate-blacks/

ple at the "Area 51" military airbase in Sept 2019. [13] The sources use the claim's photo along with other similar images showing the apparently same event. According to the VERITE annotation, the photo shows a completely different event, contradicting the evidence. Consequently, the provided labels for both claims are clearly "switched." Since we analyzed only 20 samples, there are likely more such wrongly annotated samples, penalizing DEFAME's accuracy where it should not be. Hence, the actual accuracy of DEFAME is slightly higher than measured.

## J. MOCHEG Metric Clarification

**Equivalence of Micro-F1 and Accuracy in Multi-Class Settings.** In MOCHEG's single-label, multi-class setting, micro-F1 score and accuracy are mathematically equivalent.

To see this, recall that the micro-F1 score is defined as

$$\mathrm{F1}_{\mathrm{micro}} := \frac{2 \cdot \mathrm{precision} \cdot \mathrm{recall}}{\mathrm{precision} + \mathrm{recall}} = \frac{2 \cdot \mathrm{TP}}{2 \cdot \mathrm{TP} + \mathrm{FP} + \mathrm{FN}}.$$

Now observe that since each incorrect prediction contributes exactly one false positive and one false negative, and all correct predictions are true positives, we can write

$$\mathrm{TP} + \frac{1}{2}(\mathrm{FP} + \mathrm{FN}) = \text{Total \# of predictions.}$$

Therefore, the micro-F1 score becomes

$$\mathrm{F1}_{\mathrm{micro}} = \frac{\mathrm{TP}}{\mathrm{TP} + \frac{1}{2}(\mathrm{FP} + \mathrm{FN})} = \frac{\text{Correct pred.}}{\text{All pred.}} = \text{Accuracy.}$$

This identity holds in any multiclass, single-label classification setting. We include this clarification to prevent confusion caused by legacy naming conventions in prior work on MOCHEG.

## K. Outdated Ground Truths and Temporal Dependence in MOCHEG

A qualitative analysis of failure cases in the MOCHEG dataset reveals that some ground truth explanations are no longer accurate or up-to-date, potentially affecting model evaluations (see Section 5.7). This issue arises when real-world developments render previously valid ground truths obsolete.

For instance, consider the claim:

> "A company is developing a lab-grown chicken nugget made from feathers."

**Claim # 753**
*"An image of Supreme Court Justice Clarence Thomas with his wife, Ginni Thomas, holding a bottle of wine was captured **in September 2018** after hiring Crystal Clanton to assist her media ventures."*

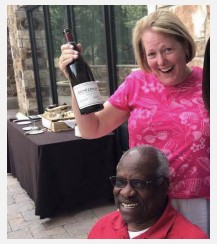

**Annotation**
✅ Supported

**Actually**
❌ OOC

**Claim # 250**
*"Image shows a crowd of people at the 'Area 51 Raid' in September 2019."*

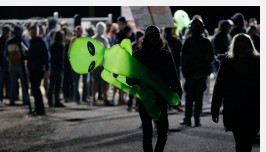

**Annotation**
❌ OOC

**Actually**
✅ Supported

**Claim # 249**
*"Image shows a crowd of people during a 'crusade' by the religious group 'In His Name Ministries' in November 2014 in Nikomazi, South Africa."*

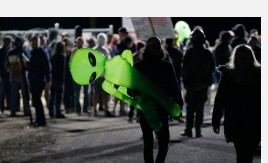

**Annotation**
✅ Supported

**Actually**
❌ OOC

Figure 9: Three faulty VERITE instances identified during qualitative analysis.

This was classified as *Not Enough Information* by Snopes, with the explanation that it is undetermined when these lab-grown nuggets will hit store shelves.[14] However, since the creation of the benchmark, lab-grown meat has become commercially available in parts of the world.[15]

Other examples of claims with temporal dependence include:

- **Claim:** "Al Gore's residence uses considerably more energy than the average American home."

- **Claim:** "Oreos have a new Lady Gaga-themed cookie."

- **Claim:** "Says Anthony Fauci will make millions off new book."

Such claims rely on a specific temporal context that may no longer be accurate as time progresses. Including such claims without temporal markers risks introducing inaccuracies into evaluation. To address the challenge of temporal dependence in fact-checking benchmarks, we propose the following alternatives:

1. **Include Timestamps:** Ensure that datasets include clear timestamps for both the claim and the associated ground truth explanation, allowing systems to account for the temporal context.

2. **Filter Out Time-Sensitive Claims:** Exclude claims with high temporal sensitivity from the dataset to avoid potential inconsistencies over time.

3. **Periodic Updates:** Regularly update benchmarks to reflect evolving ground truths, ensuring their continued relevance.

4. **Temporal Validity Check:** Integrate a pre-processing step to verify whether the ground truth explanations remain consistent with current knowledge before evaluation.

## L. Details on the Human Evaluation

We ensure a minimum number of $5$ evaluations per claim and include random samples from the Chain-of-Thought baseline for comparison. (We post-process the baseline outputs to match the format of DEFAME's output to "disguise" it.) In total, $154$ of the submissions assess the outputs from DEFAME while $31$ assess the baseline. An example of the report format is included in Appendix N. We assess the difference in Completeness scores between the DEFAME

---

[14]https://www.snopes.com/fact-check/chicken-nuggets-feather-cells

[15]https://www.theguardian.com/food/2020/dec/07/lab-grown-chicken-tastes-like-chicken-but-the-feeling-when-eating-it-is-more-complicated

and CoT LLM groups using the Mann-Whitney U Test. This non-parametric test was chosen due to the non-normal distribution of completeness scores. The Mann-Whitney U Test yields a $p$-value of approximately $9.1 \times 10^{-9}$, deeming the findings statistically significant. All Participants have a higher degree in education.

To provide further insights, we include a direct comparison of a fact-checking report generated by DEFAME and one from the Chain-of-Thought baseline for the same claim (see Figures 10 and 11). This example illustrates the key differences observed during the evaluation, particularly in the Completeness dimension. While both reports maintain coherence in structure and logical flow, the DEFAME report explicitly links its verdict to multiple pieces of evidence, providing a clear justification. In contrast, the CoT report relies heavily on parametric reasoning, lacking grounded evidence to support its conclusions.

While our human evaluation highlights DEFAME's strengths, we acknowledge certain limitations, such as the relatively small number of claims evaluated.

## M. Automatic Justification Evaluation

We also compare the generated justifications on the MOCHEG dataset to the gold rulings in Table 8. Note that our competitors here are fine-tuned on the gold rulings (except the top row and CoT). The vanilla zero-shot performance of DEFAME trails behind the other methods. We experimented by providing a single in-context example to convey the expected style of the rulings. It leads to a significant improvement in all metrics, even compared to the state-of-the-art fine-tuned models. Nonetheless, rather than speaking for much improved explanatory power, this shows the shortcomings of the automatic metrics, e.g., their sensitivity to specific n-grams and wording.

| Model | ROUGE L | BLEU | BERTScore |
|---|---|---|---|
| Best w/o FT (Yao et al., 2023) | 17.18 | 7.38 | 83.95 |
| FT (Chen et al., 2024) | 24.60 | **11.40** | **88.10** |
| FT (Yao et al., 2023) | 24.83 | 10.08 | 86.95 |
| DEFAME | 18.72 | 3.20 | 85.89 |
| DEFAME 1-shot | **25.37** | 7.31 | 87.42 |

Table 8: Performance Comparison of Explanation Generation (in %). Best scores are marked in bold, second best are underlined.

## N. Example Fact-Checking Reports

Figures 12 to 15 display two fact-checking reports as returned by DEFAME, including a correct veracity prediction. We rendered the Markdown output into a PDF, including the referenced images. These and all following reports also

## Claim

"A photograph of a riot police officer with his boot on the shoulder of a protester in 2010 during protests over a G20 summit in Toronto."

## Actions

```
reverse_search(<image:1732>)
web_search("G20 summit Toronto 2010 police protest image")
```

## Evidence

### Evidence from `reverse_search`

From [Source](): The image in question is confirmed to be from the 2010 G20 summit protests in Toronto, where police clashed with protesters. The Toronto Star reports on a $16.5 million settlement between the Toronto police board and over a thousand individuals arrested during the event, highlighting the image of police and protesters. [Socialist Action]() discusses the settlement and criticizes ongoing police misconduct, using the same image. [Analysis News]() also confirms the image's origin, discussing the aggressive police tactics and civil rights violations during the protests

### Evidence from `web_search`

From [Source](): Content: The Getty Images page contains multiple images related to the G20 summit protests in Toronto in 2010, showing police and protesters clashing. There are no specific details about the image in question, but the context aligns with the claim of police-protester interactions during the summit.

## Elaboration

The evidence confirms that the image is from the 2010 G20 summit protests in Toronto. Multiple sources, including the Toronto Star, Socialist Action, and The Analysis News, verify the context of the image, showing police clashing with protesters. The Toronto Star article mentions a settlement related to mass arrests during the event, further supporting the claim's context. Additionally, Getty Images provides a collection of similar images from the protests, reinforcing the setting and time frame.

Therefore, the claim about the photograph depicting a riot police officer with his boot on the shoulder of a protester during the 2010 G20 summit in Toronto is verified as accurate.

## Final Judgement

The fact-check confirms that the image is indeed from the 2010 G20 summit protests in Toronto. Multiple sources, including the Toronto Star and Socialist Action, verify the context of the image, showing police clashing with protesters. The Toronto Star article mentions a settlement related to mass arrests during the event, further supporting the claim's context. Getty Images also provides a collection of similar images from the protests, reinforcing the setting and time frame. The claim accurately describes the origin, content, and context of the image, as verified by multiple sources. Therefore, the most appropriate decision option is `supported`.

### Verdict: SUPPORTED

### Justification

The image is confirmed to be from the 2010 G20 summit protests in Toronto, depicting police clashing with protesters. Multiple sources, including the [Toronto Star,]() [Socialist Action]() and [The Analysis News]() verify the context and authenticity of the image. The Toronto Star article also mentions a settlement related to mass arrests during the event, supporting the claim's context. Getty Images provides additional images from the protests, reinforcing the setting and time frame. The claim is thus supported.

Figure 10: Fact-checking report by DEFAME, presented in the Human Evaluation.

## Claim

"A photograph of a riot police officer with his boot on the shoulder of a protester in 2010 during protests over a G20 summit in Toronto."

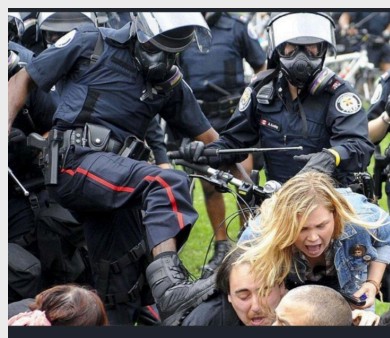

## Final Judgement

The image shows a police officer with his boot on the shoulder of a protester, which aligns with the claim that it was taken during protests over a G20 summit in Toronto in 2010.

This event was marked by significant clashes between police and protesters, and the image accurately reflects the tense and confrontational atmosphere of those protests.

Therefore, the claim accurately describes the origin, content, and context of the image. The decision option that applies best is `supported`.

Verdict: SUPPORTED

### Justification

The claim that the image depicts a police officer with his boot on a protester's shoulder during the 2010 G20 summit protests in Toronto is supported. The image accurately reflects the confrontational atmosphere of the event, aligning with the described context and content.

Figure 11: Fact-checking report by the Chain-of-Thought baseline, presented in the Human Evaluation.

include hyperlinks[16], referencing the used resources.

## O. Examples of Failure Cases

The qualitative analysis of mispredicted VERITE instances uncovered two common failure modes attributed to DE-FAME: *premature judgment* and *failed evidence retrieval*. See Figure 16 for an example of a premature judgment. While it is true that there exists a photo showing the "Air Force Officers' Spouses' Club" taken on April 24, 2018[17], it is not the photo depicted in the claim. DEFAME missed comparing the claim photo with evidence photos before judging the veracity.

Additionally, Figures 17 and 18 show a case where the retrieval of evidence (from either (reverse) image search or web search) was unsuccessful, resulting in a wrong prediction. Manual inspection reveals that according to Snopes[18], the origin is a news video accessible on YouTube. However, both Snopes and YouTube are excluded from DEFAME's search results. Apart from that, manually using all three web search tools yields only more excluded or unrelated results.

---

[16]For technical reasons, the hyperlinks are not preserved in this PDF.

[17]https://web.archive.org/web/20241116170848/https://taskandpurpose.com/military-life/viral-military-spouse-tweet-photo/

[18]https://www.snopes.com/fact-check/sharks-on-power-lines-hurricane/

## Claim

"A drunken man in a basket. Bars in Turkey employed basket man who's job it was to take patrons home who were too drunk to stand up. Istanbul 1960s."

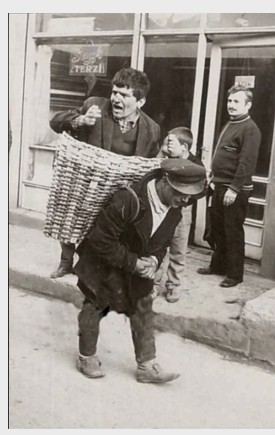

## Actions

```
geolocate()
web_search("basket man Istanbul 1960s Turkey")
image_search("basket man Istanbul 1960s Turkey")
```

## Evidence

### Evidence from `geolocate`

The most likely countries where the image was taken are: {'Turkey': 0.26, 'Albania': 0.22, 'Bulgaria': 0.18, 'Serbia': 0.06, 'Hungary': 0.05, 'Palestine': 0.05, 'Montenegro': 0.04, 'Macedonia': 0.03, 'Portugal': 0.02, 'Andorra': 0.01}

### Evidence from `image_search`

The concept of "basket men" in Istanbul during the 1960s involved individuals employed to carry home patrons who were too drunk to stand. An image from Paintvine shows a man carrying another in a basket, similar to the one described in the claim, but without additional context. Paintvine History in Memes provides further discussion on this practice, confirming its existence during that time period.

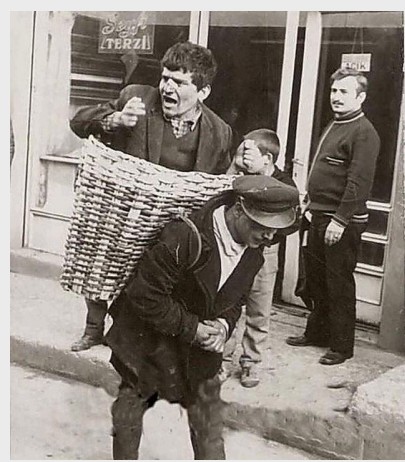 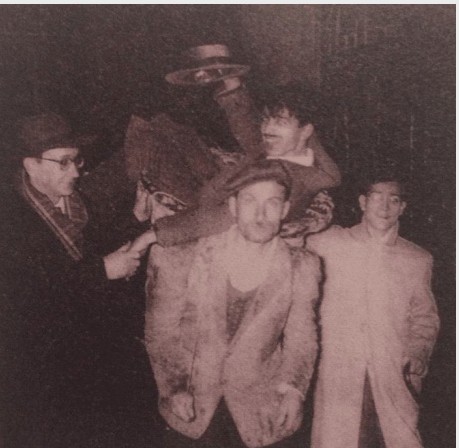

Figure 12: Exemplary fact-check report with a correct prediction, page 1 of 2.

## Elaboration

The evidence supports the claim that "basket men" were employed in Istanbul during the 1960s to carry home patrons who were too drunk to stand. The image search confirms the existence of this practice, showing similar images and providing context about the role of these individuals. The geolocation analysis suggests Turkey as a likely location for the image, aligning with the claim's mention of Istanbul. However, the specific details about the individuals in the image, such as whether the person in the basket is indeed drunk, cannot be confirmed solely from the evidence provided. Overall, the claim about the practice and its location is supported by the evidence.

## Final Judgement

The fact-check reveals that the practice of employing "basket men" in Istanbul during the 1960s to carry home patrons who were too drunk to stand is supported by historical context and similar images. The geolocation analysis also suggests Turkey as a likely location, aligning with the claim. However, specific details about the individuals in the image, such as the state of the person in the basket, cannot be confirmed.

Given the evidence, the claim about the practice and its location is supported. Therefore, the most appropriate decision option is `supported`.

Verdict: SUPPORTED

Justification

The fact-check confirms that the practice of employing "basket men" in Istanbul during the 1960s to carry home patrons who were too drunk to stand is supported by historical context and similar images. Geolocation analysis suggests Turkey as a likely location, aligning with the claim. However, specific details about the individuals in the image, such as the state of the person in the basket, cannot be confirmed. Overall, the claim about the practice and its location is supported. Paintvine History in Memes.

Figure 13: Exemplary fact-check report with a correct prediction, page 2 of 2.

## Claim

"Image of Joe Biden kissing his 19-year-old granddaughter, Finnegan Biden."

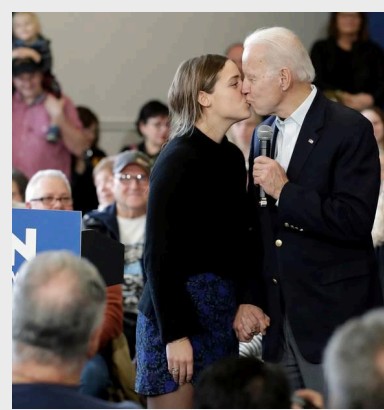

## Actions

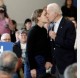

```
reverse_search(<image:1328>)
web_search("Joe Biden kissing granddaughter Finnegan Biden")
```

## Evidence

### Evidence from `reverse_search`

From [Source](): Found exact image on website with title: Joe Biden kisses granddaughter on lips during Iowa rally Content: The New York Post reports that Joe Biden kissed his 19-year-old granddaughter, Finnegan Biden, on the lips during a campaign rally in Iowa. The event took place at Clarke University in Dubuque. The article includes images of the kiss:

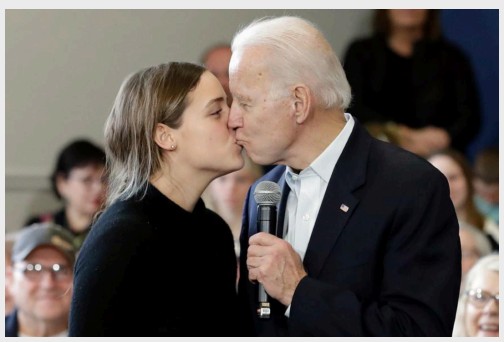
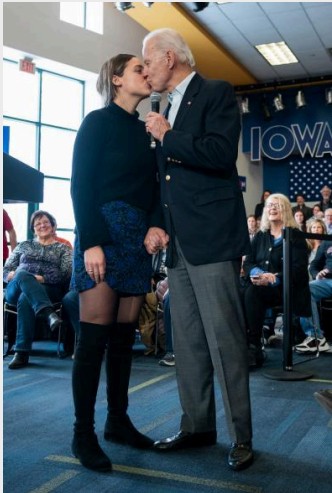

The incident prompted mixed reactions on social media. The article also mentions past criticisms of Biden's behavior with women.

Figure 14: Exemplary fact-check report with a correct prediction, page 1 of 2.

Evidence from `web_search`

The photo in question shows Joe Biden kissing his 19-year-old granddaughter, Finnegan Biden, during a campaign stop in Iowa. This event took place in February at Clarke University in Dubuque. The claim that he was kissing a 15-year-old is false. The gesture sparked mixed reactions, with some finding it unsettling and others viewing it as a normal display of affection.

For more details, you can refer to the articles from AP News and NZ Herald.

## Elaboration

The evidence confirms that the image shows Joe Biden kissing his granddaughter, Finnegan Biden, during a campaign rally in Iowa. The event took place at Clarke University in Dubuque. The New York Post and other sources, such as AP News and NZ Herald, corroborate this information. The claim that she is 19 years old is also supported by these sources. Therefore, the claim is verified as true.

## Final Judgement

The fact-check confirms that the image shows Joe Biden kissing his granddaughter, Finnegan Biden, during a campaign rally in Iowa. The event took place at Clarke University in Dubuque, and multiple sources, including the New York Post, AP News, and NZ Herald, verify this information. The claim that she is 19 years old is also supported by these sources, confirming the accuracy of the claim.

The caption is factually correct, and the image aligns with the claim. There is no indication that the image is used out of context, as it accurately represents the described event. Therefore, the claim is supported by the evidence provided.

`supported`

Verdict: SUPPORTED

Justification

The fact-check confirms that the image shows Joe Biden kissing his granddaughter, Finnegan Biden, during a campaign rally at Clarke University in Dubuque, Iowa. Multiple sources, including the New York Post, AP News, and NZ Herald, verify this information and confirm that she is 19 years old. The claim is accurate and supported by the evidence.

Figure 15: Exemplary fact-check report with a correct prediction, page 2 of 2.

## Claim

"Image depicts members of the Air Force Officers' Spouses' Club taken at the annual Joint Armed Forces of Washington Luncheon (JAFOWL) on April 24."

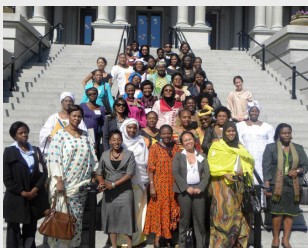

## Actions

```
reverse_search(<image:1551>)
web_search("Air Force Officers' Spouses' Club annual Joint Armed Forces
          of Washington Luncheon April 24")
```

## Evidence

### Evidence from `web_search`

From [Source](): Content: The photo in question was taken at the annual Joint Armed Forces of Washington Luncheon (JAFOWL) on April 24, not at the White House. It depicts members of the Air Force Officers' Spouses' Club. The event is criticized for potentially reinforcing a hierarchy among military spouses. The image was mistakenly associated with a White House event by Ronald Klain in a viral tweet.

## Elaboration

The evidence confirms that the image was taken at the annual Joint Armed Forces of Washington Luncheon (JAFOWL) on April 24 and depicts members of the Air Force Officers' Spouses' Club. This aligns with the claim, verifying its accuracy. The image was mistakenly associated with a White House event in a viral tweet, but this does not affect the claim's validity regarding the event and group depicted. Therefore, the claim is true based on the provided evidence.

## Final Judgement

The fact-check confirms that the image was taken at the annual Joint Armed Forces of Washington Luncheon (JAFOWL) on April 24 and depicts members of the Air Force Officers' Spouses' Club. This aligns with the claim, verifying its accuracy. Although the image was mistakenly associated with a White House event in a viral tweet, this does not affect the claim's validity regarding the event and group depicted.

The claim accurately describes the origin, content, and context of the image. Therefore, the most appropriate decision option is `supported`.

### Verdict: SUPPORTED

### Justification

The fact-check confirms that the image was taken at the annual Joint Armed Forces of Washington Luncheon (JAFOWL) on April 24 and depicts members of the Air Force Officers' Spouses' Club, as claimed. Despite a viral tweet mistakenly associating the image with a White House event, the evidence from [Task & Purpose]() supports the claim's accuracy regarding the event and group depicted. Therefore, the claim is supported.

Figure 16: Report of a fact-check which ended in a wrong prediction due to a premature judgment.

## Claim

"Image shows portions of a fishnet as well as several fish being stuck up on power lines after traffic accident with a fisherman boat, in August 2020."

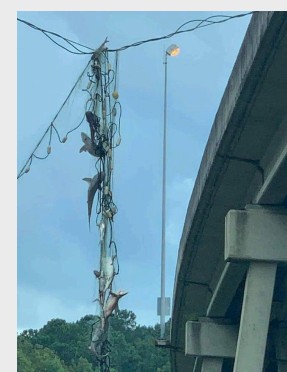

## Actions

```
reverse_search(<image:1312>)
web_search("fishnet power lines traffic accident fisherman boat August 2020")
```

## Evidence

No new evidence found.

## Elaboration

The claim suggests that a traffic accident involving a fisherman's boat resulted in a fishnet and several fish being stuck on power lines. However, there is no new evidence to support or refute this claim. The image alone does not provide enough context or details to verify the incident or its cause.

To assess the claim's veracity, additional information is needed, such as news reports or eyewitness accounts from August 2020 that describe a similar incident. Without such evidence, the claim remains unverified.

## Actions

```
image_search("fishnet power lines traffic accident August 2020")
web_search("fishnet power lines accident fisherman boat August 2020")
```

## Evidence

No new evidence found.

## Elaboration

The claim remains unverified due to a lack of supporting evidence. The image alone does not provide sufficient context to confirm the incident described. No additional information, such as news reports or eyewitness accounts from August 2020, has been found to substantiate the claim. Therefore, without further evidence, the claim cannot be confirmed or refuted.

Figure 17: Report of a fact-check (page 1 of 2), which ended in a wrong prediction due to the failed retrieval of evidence.

## Actions

```
image_search("fishnet power lines accident August 2020")
web_search("fishnet power lines traffic accident August
2020")
```

## Evidence

No new evidence found.

## Elaboration

The claim remains unverified due to insufficient evidence. Despite multiple searches, no supporting information such as news articles or eyewitness accounts from August 2020 has been found to confirm the incident described. The image alone lacks context and details necessary to substantiate the claim. Without additional evidence, the claim cannot be confirmed or refuted. Further investigation or credible sources are needed to verify the event.

## Final Judgement

The claim suggests that a traffic accident involving a fisherman's boat resulted in a fishnet and several fish being stuck on power lines. Despite multiple searches, no supporting evidence such as news reports or eyewitness accounts from August 2020 has been found to confirm this incident. The image alone does not provide enough context or details to verify the claim.

Given the lack of evidence and context, the claim appears to construct a narrative around the image without substantiation. Therefore, the claim is best categorized as `out of context`.

Verdict: OUT_OF_CONTEXT

Justification

The claim that a traffic accident involving a fisherman's boat resulted in a fishnet and several fish being stuck on power lines remains unverified due to a lack of supporting evidence. Despite multiple searches, no news reports or eyewitness accounts from August 2020 have been found to confirm the incident. The image alone lacks sufficient context to substantiate the claim, leading to the conclusion that the claim is `out of context`.

Figure 18: Report of a fact-check (page 2 of 2), which ended in a wrong prediction due to the failed retrieval of evidence.

