# OpenReview forum: "DEFAME: Dynamic Evidence-based FAct-checking with Multimodal Experts"
_ICML.cc/2025/Conference — ICML 2025 poster_

### Official Review · Reviewer_3bbT · 2025-03-14

**Overall Recommendation:** 2

**Summary:**

The authors introduce DEFAME, an automated fact-checking framework designed to process multimodal claims using multimodal evidence. DEFAME operates within a zero-shot MLLM pipeline structured into six stages: action planning, action execution (via multimodal web retrieval and GeoClip tool use), result summarization, reasoning about claim veracity, verdict prediction, and verdict justification. Evaluated on three established multimodal fact-checking datasets (AVERITEC, MOCHEG, VERITE) and a newly proposed dataset (ClaimReview2024+), DEFAME outperforms existing MLLMs and multimodal fact-checking approaches while generating higher-quality fact-checking reports compared to the base MLLM with CoT prompting.

**Claims And Evidence:**

Certain claims made in the submission lack sufficient supporting evidence:

1. Novelty of DEFAME: The authors claim that DEFAME is "the first multimodal AFC system that can handle multimodal claims as well as retrieve and process multimodal evidence" (Lines 72–74). However, this is fundamentally inaccurate, as prior work [1] has already implemented multimodal retrieval (both text and image) for multimodal misinformation detection and was evaluated on the same MOCHEG dataset. This undermines the claimed novelty of DEFAME.

2. State-of-the-art results: The claim that DEFAME "establishes new state-of-the-art results on three diverse and widely used benchmarks" (Lines 78-80) is only partially supported by Table 3. The table presents incomplete comparisons, as only GPT-4o and GPT-4o with CoT are evaluated on all four datasets, while other baselines are tested on at most two datasets. This weakens the claim that DEFAME definitively outperforms prior approaches.


[1] Tahmasebi et al., Multimodal Misinformation Detection using Large Vision-Language Models, arXiv:2407.14321, 2024. (Published at CIKM 2024)

**Essential References Not Discussed:**

Please refer to References [1] and [2] under "Relation To Broader Scientific Literature".

Additionally, the authors fail to discuss SNIFFER [3], a representative work that leverages MLLMs for explainable out-of-context misinformation detection. Given SNIFFER’s focus on MLLM-empowered explainable detection, its omission further weakens the discussion of related work.

[3] Qi et al., SNIFFER: Multimodal Large Language Model for Explainable Out-of-Context Misinformation Detection. CVPR 2024.

**Experimental Designs Or Analyses:**

While the authors claim that DEFAME achieves state-of-the-art performance across diverse benchmarks, the experimental results in Table 3 are incomplete. Specifically, apart from GPT-4o and GPT-4o with CoT, all other baselines are evaluated on at most two out of the four datasets, limiting the robustness of the comparison. Furthermore, on the newly introduced ClaimReview2024+ dataset, DEFAME is only compared against base MLLM approaches that lack task-specific framework design, web retrieval, and tool use, making its superiority expected. These gaps weaken the claim that DEFAME establishes a new state of the art.

Additionally, Table 7 reveals significant efficiency concerns. DEFAME with GPT-4o requires around 28× the time and 21× the input tokens compared to GPT-4o with CoT, and even its ablated variants remain highly resource-intensive. This raises serious questions about DEFAME’s practicality for real-world deployment.

**Methods And Evaluation Criteria:**

The authors evaluate DEFAME on four datasets (three existing and one newly introduced), which represents a comprehensive selection. However, the inconsistency in evaluation metrics across datasets is not justified. Specifically, while Accuracy is reported for most datasets, F1 score is used for MOCHEG, without explanation.

**Other Comments Or Suggestions:**

N/A

**Other Strengths And Weaknesses:**

Please refer to comments under "Relation To Broader Scientific Literature", "Methods And Evaluation Criteria", and "Experimental Designs Or Analyses".

**Questions For Authors:**

1. Completion of Table 3: The performance of existing approaches beyond GPT-4o and GPT-4o with CoT remains unclear across all four datasets, and the evaluation metrics are inconsistent (Accuracy vs. F1 score). Specifically, how do retrieval-augmented approaches perform on the newly constructed ClaimReview2024+ dataset compared to DEFAME? The lack of such comparisons weakens the claim of DEFAME’s superiority.

2. Efficiency of Existing MLLM-Empowered Approaches: Table 7 shows that DEFAME consumes significantly more tokens and execution time than MLLM prompting, which is expected due to its evidence retrieval process. However, there is no direct comparison with other task-specific MLLM-empowered approaches that also utilize evidence retrieval. Without this, it remains unclear how DEFAME’s efficiency compares to competing baselines.

3. Explanation Quality Comparison: In Section 4.6, DEFAME’s explanation quality is only compared against bare MLLM prompting (GPT-4o with CoT), which lacks external evidence access. Given this, DEFAME’s superior performance is expected rather than insightful. A missing analysis is how DEFAME’s generated explanations compare to those from other retrieval-augmented, task-specific MLLM approaches.

**Relation To Broader Scientific Literature:**

Compared to the broader scientific literature, the conceptual and technical contributions of this paper remain highly limited.

The claimed key novelty of DEFAME -- being "the first multimodal AFC system that can handle multimodal claims as well as retrieve and process multimodal evidence" (Lines 72–74) -- is inaccurate. Prior work [1] has already implemented multimodal evidence retrieval (text and image) for misinformation detection and was evaluated on the same MOCHEG dataset.

DEFAME is a zero-shot MLLM pipeline for standard multimodal fact-checking, but its retrieval paradigm closely resembles existing retrieval-augmented MLLM approaches, such as [1] and MMD-Agent [2]. While DEFAME's integration of reverse image search and geolocation is novel in execution, these contributions are incremental given the lack of originality in problem formulation and overall framework design.

[1] Tahmasebi et al., Multimodal Misinformation Detection using Large Vision-Language Models, arXiv:2407.14321, 2024. (Published at CIKM 2024)

[2] Liu et al., MMFakeBench: A Mixed-Source Multimodal Misinformation Detection Benchmark for LVLMs. arXiv: 2406.08772, 2024. (Published at ICLR 2025)

**Theoretical Claims:**

N/A: the paper does not contain theoretical claims.

---

> ### Author Rebuttal · Authors · 2025-03-31
>
> We thank for the time invested into the review. Please find our response to your concerns below.
> ### First to handle multimodal claims and evidence
> The work by Tahmasebi et al. (CIKM 2024), i.e. LVLM4F, was covered in our paper, most notably in the prior work overview (Table 1) and as a baseline in Table 3. Critically, LVLM4FV **is not designed to handle multimodal claims**: “Given a **textual** claim 𝑄 and a corpus of evidences C as input [...]”, p. 3.
> ### Key novelty
> Please refer to our response to Reviewer bvvH (“Originality and Theoretical Contribution”) for more details on the novelty of DEFAME.
> ### Prior work
> Thanks for pointing out SNIFFER. Please refer to our response to Reviewer K7Ta (“Comparison to Prior Work”).
> ### ”Incomplete” Table 3
> The other methods in Table 3 have not been evaluated on all benchmarks for several methodological reasons. Most importantly, the methods have different task specializations, targeting only a particular subtask of fact-checking: CFR, GPT-CoT, LVLM4FV, and MetaSum are limited to text-only claims (not provided by VERITE and partially CR+). CHASMA and AITR focus solely on OOC detection and both require visual input, making them inapplicable to text-only claims (like in AVeriTeC, MOCHEG, and partially CR+). For CFR, the code and model weights have not been publicly released. GPT-CoT requires gold evidence as input to achieve the reported competitive numbers, not available for VERITE and CR+, where evidence retrieval is considered an integral part of the fact-checking task. Moreover, GPT-CoT mainly builds on GPT-3.5-Turbo, implying that the much stronger GPT-4o (CoT) baselines provide a better comparison than GPT-CoT. CHASMA lacks a publicly available, trained model. Furthermore, CHASMA and AITR both require training - which is impossible for CR+ as there is no training data available. LVLM4FV, MetaSum, and AITR require a predefined evidence corpus for retrieval, which is infeasible to create for CR+.
>
> Thus, it is not possible to run the previous methods on the benchmarks they were not designed for (or if it is technically feasible, it will produce meaningless results). With DEFAME, we deliver a method that - despite its generality - is able to beat even the specialized methods.
> ### Inconsistency in metrics
> Metrics (F1, accuracy) match those used in the original benchmarks and prior work for consistent intra-benchmark comparison. If the reviewer finds it helpful, we are open to adding F1 score as an additional metric to AVeriTeC and CR+ or, alternatively, add accuracy to MOCHEG.
> ### Efficiency
> We agree that DEFAME has high token usage and acknowledge that there is room to reduce it. As correctly anticipated, the high token consumption is due to the processing of external evidence. External evidence can incorporate entire webpages, full PDFs, and other long documents that Firecrawl can turn into Markdown representation. We truncate inputs only when they exceed the maximum context window of the MLLM, which is 128k tokens for the GPT models used. Since the GPT baselines do not process any external evidence, unsurprisingly, their token consumption is fairly low - *at the cost of random-like performance for unseen claims*. Table 7 also reveals that the integration of planning (DEFAME with GPT-4o) reduces token consumption by about 20K tokens (almost a third) compared to the planning-ablated variant “Static Actions,” which executes all available actions. Since efficiency was not a goal of DEFAME, we offer to add it as a future direction to the discussion section.
>
> It is hard to compare DEFAME’s efficiency with the methods in Table 3 due to (a) no reported resource usage (e.g., tokens or FLOPs), and (b) significant differences in architecture; also many are specialized for sub-tasks like OOC detection or evidence summarization and do not perform retrieval. Thus, direct efficiency comparison is not meaningful.

---

> > ### Comment · Reviewer_3bbT · 2025-04-05
> >
> > I thank the authors for their clarification regarding LVLM4F and, accordingly, I have raised my score to 2.
> >
> > I have also reviewed the authors' response to Reviewer K7Ta regarding the “Comparison to Prior Work.” To clarify, my concern, shared by other reviewers, is not about the novelty of this work per se. Rather, the omission of closely related works such as SNIFFER and MMFakeBench hinders a comprehensive understanding of the current landscape in multimodal automated fact-checking. Discussing these works is important for properly contextualizing the contributions of this paper.
> >
> > Finally, I recommend aligning the evaluation metrics across datasets (i.e., reporting both Accuracy and F1 scores consistently). Otherwise, please include a clear explanation in the experimental section for only reporting Accuracy or F1 for certain datasets.

---

> > > ### Author Response · Authors · 2025-04-08
> > >
> > > We thank the reviewer for their constructive feedback and appreciate the opportunity to clarify the evaluation metrics and our handling of related work.
> > >
> > > ### On Related Work
> > > We acknowledge the omission of SNIFFER and have addressed it in our response, along with detailed distinctions between DEFAME, SNIFFER, and MMFakeBench (MMD-Agent). These clarifications will be incorporated into the final version. If the reviewer has additional suggestions, we are happy to include them as well.
> > > Most importantly, **none of these works renders DEFAME obsolete**. To the best of our knowledge, DEFAME remains the only system that jointly supports multimodal claims and multimodal evidence, dynamic tool planning, and full explanatory output in a zero-shot setting.
> > > ### On Evaluation Metrics (Accuracy vs. F1)
> > > MOCHEG is the only benchmark where we report micro-F1 score, following the convention established in its original paper. All the other benchmarks use accuracy.
> > >
> > > Note that **micro-F1 and accuracy are mathematically equivalent** in this setting—multiclass classification with exactly one correct label per instance. The reason for this confusion/redundancy stems from the MOCHEG paper. Close examination of the MOCHEG codebase (and follow-up works) reveals that the metric uses the standard [scikit-learn](https://scikit-learn.org/stable/modules/generated/sklearn.metrics.f1_score.html) implementation of micro-F1 score, which [indeed is](https://scikit-learn.org/stable/modules/model_evaluation.html#multiclass-and-multilabel-classification) accuracy in the multi-class-single-label setting. **We will add this explanation to the final version for clarity.**
> > >
> > > We provide a complementary explanation below to clarify the equivalence of micro F1 and accuracy in the multi-class-single-label setting:
> > >
> > > This is due to two key properties:
> > >
> > > 1. **No True Negatives (TN):** In multiclass prediction, each instance has a single gold label. Therefore, every correct prediction is a True Positive for a specific class. Hence, there are no True Negatives; each prediction is either a True Positive or an incorrect prediction.
> > > 2. **One-to-one error symmetry:** Every incorrect prediction contributes **one False Positive and one False Negative**, implying FP = FN and total false predictions are double-counted in standard F1 terms.
> > >
> > > Micro F1 is defined as
> > > $$
> > > \text{F1}_{\text{micro}} := \frac{2 \cdot \text{precision} \cdot \text{recall}}{\text{precision} + \text{recall}} = \frac{2 \cdot \text{TP}}{2 \cdot \text{TP} + \text{FP} + \text{FN}}.
> > > $$
> > >
> > > Now observe that since the TPs capture all correct predictions and since each incorrect prediction contributes one FP and one FN, we can write:
> > > $$
> > > \text{TP} + \frac{1}{2}(\text{FP} + \text{FN}) = \text{All predictions}.
> > > $$
> > >
> > > Therefore,
> > > $$
> > > \text{F1}_{\text{micro}} = \frac{\text{TP}}{\text{TP} + \frac{1}{2}(\text{FP} + \text{FN})} = \frac{\text{Correct predictions}}{\text{All predictions}} = \text{Accuracy}.
> > > $$
> > >
> > > This identity holds in any multiclass single-label setting, and we will make this equivalence explicit in the paper to avoid confusion.

---

### Official Review · Reviewer_bvvH · 2025-03-16

**Overall Recommendation:** 2

**Summary:**

This paper tackles the challenge of scalable and explainable fact-checking in the presence of disinformation, particularly in multimodal contexts. The authors propose DEFAME, a modular, zero-shot multimodal large language model (MLLM) pipeline for open-domain claim verification. Unlike prior methods that are either text-only or overly reliant on parametric knowledge, DEFAME operates through a six-stage dynamic tool selection process, incorporating both textual and visual evidence to generate structured, explainable verification reports. Extensive evaluation on VERITE, AVERITEC, and MOCHEG demonstrates DEFAME’s superiority over existing fact-checking models, setting a new state-of-the-art for uni- and multimodal fact-checking. Furthermore, the authors introduce CLAIMREVIEW2024+, a new benchmark that ensures post-GPT-4O knowledge cutoff validity, highlighting DEFAME’s temporal generalizability and real-time fact-checking capabilities, significantly outperforming the GPT-4O Chain-of-Thought baseline.

**Claims And Evidence:**

Claim: The paper claims that DEFAME is the first multimodal Automated Fact-Checking (AFC) system capable of handling multimodal claims while also retrieving and processing multimodal evidence.
Question: However, based on my literature review using Google Scholar, there are existing studies [r1], [r2], [r3] that employ Large Vision-Language Models (LVLMs) with knowledge retrieval for multimodal misinformation detection.

[r1] Qi P, et al. SNIFFER: Multimodal Large Language Model for Explainable Out-of-Context Misinformation Detection. CVPR, 2024.
[r2] Liu X, et al. Mmfakebench: A mixed-source multimodal misinformation detection benchmark for lvlms. ICLR 2025.
[r3] Xuan K, et al. LEMMA: Towards LVLM-Enhanced Multimodal Misinformation Detection with External Knowledge Augmentation. 2024.

**Essential References Not Discussed:**

Two key related works—Sniffer (r1 SNIFFER: Multimodal Large Language Model for Explainable Out-of-Context Misinformation Detection. CVPR, 2024.) and MMFakeBench (r2 Mmfakebench: A mixed-source multimodal misinformation detection benchmark for lvlms. ICLR 2025.)—are missing from the current discussion, and they are essential for understanding the context of multimodal fact-checking and misinformation detection.

**Experimental Designs Or Analyses:**

Strengths of Experimental Design
1.	Use of Diverse and Established Benchmarks. The evaluation leverages three well-known datasets (VERITE, AVERITEC, MOCHEG) to assess DEFAME's text, image-text, and multimodal fact-checking capabilities.
2.	Ablation Studies for Key Components. The six-stage verification process is systematically analyzed through component ablations (removal of Web Search, Image Search, Geolocation, Reverse Image Search).

Weakness of Experimental Design:
1.	CLAIMREVIEW2024+ is an important contribution, but the dataset construction methodology lacks clarity. Unclear aspects: What criteria were used for claim selection? How were fact-checking labels assigned and validated? Does it truly reflect real-world misinformation trends? A more detailed dataset creation process would improve trust and reproducibility.

**Methods And Evaluation Criteria:**

The proposed methods and evaluation criteria are well-aligned with the problem of multimodal fact-checking and effectively demonstrate the capabilities of DEFAME in real-world misinformation detection.

1.	Diverse Benchmark Selection. The authors evaluate DEFAME on text-only (AVERITEC), image-text (VERITE), and multimodal (MOCHEG) datasets, covering varied fact-checking challenges.
2.	The introduction of CLAIMREVIEW2024+ is commendable, as it tests DEFAME’s ability to verify claims beyond the GPT-4O knowledge cutoff, addressing potential data leakage issues.

**Other Comments Or Suggestions:**

Line 48 (Most multimodal claim verification systems cannot even retrieve the evidence needed to verify a claim (Fu et al., 2024; Vo-Hoang et al., 2024; Tang et al., 2024),.) contains an extraneous comma that was mistakenly added.

**Other Strengths And Weaknesses:**

The paper has notable strengths in terms of application significance, empirical rigor, and modular system design, but also has some weaknesses related to originality, clarity of contributions, and unexplored limitations.
Strengths:
1.	Strong Empirical Validation. The evaluation is conducted on three well-established AFC benchmarks (VERITE, AVERITEC, MOCHEG) and a new dataset (CLAIMREVIEW2024+), providing comprehensive empirical insights.
2.	Modular and Scalable System Design. DEFAME is modular, allowing flexibility in integrating different retrieval tools (Web Search, Image Search, Reverse Image Search, Geolocation).

Weaknesses for Improvement
1. Originality Could Be Better Clarified. While DEFAME combines existing techniques innovatively, it does not introduce a fundamentally new theoretical model. The paper does not sufficiently differentiate DEFAME’s retrieval mechanism from prior RAG-based AFC models
2. If possible, provide a more formal description of the six-stage verification pipeline to enhance clarity.
3. The mathematical formulation of DEFAME’s retrieval pipeline and reasoning steps could be clearer.
4. The dataset construction details for CLAIMREVIEW2024+ are not fully transparent.

**Questions For Authors:**

Overall, this paper presents a well-structured approach to multimodal fact-checking with strong empirical validation. However, to align with ICML’s emphasis on theoretical rigor, I would like to ask the following critical question regarding the methodological formalization in Section 3:
 Can the authors introduce a more formal theoretical and mathematical representation of the method in Section 3?
* The current presentation of DEFAME’s six-stage verification process is primarily descriptive and algorithmic, but lacks a formal mathematical framework. Incorporating a structured formalization would significantly enhance the paper’s impact.

**Relation To Broader Scientific Literature:**

The paper provides a clear review of the three key AFC components: Claim detection & extraction Evidence retrieval Verdict prediction
It references leading AFC models, such as: Text-only AFC systems (e.g., FEVER, AVERITEC, FACTCHECK-BENCH) Multimodal AFC systems (e.g., VERITE, MOCHEG, NEWSCLIPPINGS).

DEFAME extends prior work: Unlike text-only AFC systems, DEFAME incorporates multimodal evidence retrieval and reasoning. Unlike previous multimodal AFC approaches, DEFAME performs dynamic evidence retrieval rather than relying solely on pre-annotated evidence.

**Theoretical Claims:**

The paper primarily focuses on the practical application of multimodal fact-checking rather than developing new theoretical foundations or formal proofs. As such, no explicit theoretical claims or mathematical proofs are presented that require verification. Therefore, this raises a potential concern regarding the lack of formal theoretical justification for some of the proposed design choices in DEFAME.

---

> ### Author Rebuttal · Authors · 2025-03-31
>
> ### Claim on Novelty / Related Work
> Thanks for pointing out SNIFFER, MMD-Agent (from the MMFakeBench paper), and LEMMA. Please refer to our response to Reviewer K7Ta.
> ### No Theoretical Claims
> We believe that absence of theoretical claims is not unusual for papers under “Application-Driven Machine Learning”, the ICML topic that we have submitted to. At the same time, we offer a more formal mathematical representation of the framework below.
>
> ### No Supplementary Material, Details for CR+
> Please refer to Appendix Sections A–L (pp. 16 ff.), which contain extensive additional material. Some of your questions are addressed in Appendix J. We also point you to our response to Reviewer K7Ta on “ClaimReview2024+ Details”, where these details are further elaborated.
> ### Originality and Theoretical Contribution
> DEFAME’s originality lies in its dynamic, modular integration of retrieval tools for handling both multimodal claims **and** multimodal evidence—capabilities not jointly supported by prior work. While it builds on established components like CoT prompting and tool use, DEFAME combines them into a unified framework that supports zero-shot fact-checking across a diverse set of benchmarks: AVeriTeC (text claims & text evidence), MOCHEG (text claims and multimodal evidence), VERITE (multimodal claims and potentially multimodal evidence), ClaimReview2024+ (both uni- and multimodal claims with potentially multimodal evidence). To our knowledge, it is the only system capable of operating across all these scenarios, while previous methods are specialized and typically incompatible with at least one modality setting (e.g., requiring only text, only images, or static corpora). We will make this contribution more explicit.
>
> ### Formal Representation of DEFAME
>
> We appreciate the suggestion to formalize the DEFAME pipeline. We agree that a formal view helps clarify the role of each stage.
>
> Let $\mathcal{T}$ and $\mathcal{I}$ be the spaces of text and images. Define $\mathcal{M} := (\mathcal{T} \cup \mathcal{I})^*$ as the space of multimodal sequences, and $\mathcal{Y}$ the space of verdict labels. DEFAME is a function:
>
> $$
> \mathcal{F} : \mathcal{M} \rightarrow \mathcal{M} \times \mathcal{Y}, \quad \mathcal{F}(c) = (R_\text{out}, y_\text{out}),
> $$
>
> where, given a claim $c \in \mathcal{M}$, the output consists of a report $R_\text{out}$ containing the fact-check, and a predicted verdict $y_\text{out}$. DEFAME proceeds iteratively up to $N$ steps as follows:
>
> - $(R^{(i+1)}, y^{(i+1)}) := \mathcal{F}_\text{iter}(R^{(i)})$
> - $i^* := \min \\{ i \leq N \mid y^{(i)} \ne \text{NEI} \text{ or } i = N \\}$
>
> Final outputs:
>
> - $R_\text{out} := \mathcal{S}_6(R^{(i^*)})$ (justification)
> - $y_\text{out} := y^{(i^*)}$
> The justification stage $\mathcal{S}_6$ appends a rationale to the final report.
>
> Each iteration
> $$\mathcal{F}_\text{iter} = \mathcal{S}_5 \circ \mathcal{S}_4 \circ \mathcal{S}_3 \circ \mathcal{S}_2 \circ \mathcal{S}_1$$
> consists of:
> 1. Planning ($\mathcal{S}_1$): Select actions $A \subseteq \mathcal{A}$ based on $R^{(i)}$
> 2. Execution ($\mathcal{S}_2$): Retrieve evidence $E := \\{ \tau(a) \mid a \in A \\}$, where $\tau$ is a tool executing corresponding action $a$.
> 3. Summarization ($\mathcal{S}_3$): $R_1^{(i)} := \sigma(E, R^{(i)})$, where $\sigma$ summarizes evidence $E$ conditioned on the current report $R^{(i)}$ and appends it to the report.
> 4. Develop ($\mathcal{S}_4$): $R_2^{(i)} := \mathcal{S}_4(R_1^{(i)})$, where $\mathcal{S}_4$ is a generative model that performs structured reasoning and expands the report with the generated NLI sequence.
> 5. Verdict Prediction ($\mathcal{S}_5$): $(R_3^{(i)}, y^{(i)}) := \mathcal{S}_5(R_2^{(i)})$, where $\mathcal{S}_5$ is a classifier over multimodal sequences, returning a verdict $y^{(i)} \in \mathcal{Y}$ and an updated report $R_3^{(i)} \in \mathcal{M}$ which is the input report $R_2^{(i)}$ expanded by a summary of the key takeaways from the report alongside with the verdict.
>
> We welcome feedback on whether including this formalization (or an extended/abbreviated version) in the main paper would be helpful.

---

### Official Review · Reviewer_K7Ta · 2025-03-18

**Overall Recommendation:** 3

**Summary:**

The paper  presents a novel approach to automated fact-checking designed to address the growing problem of disinformation. The authors introduce DEFAME, a modular system that uses a six-stage pipeline to dynamically select and use various tools for retrieving and evaluating both textual and visual evidence. This system is capable of handling multimodal claims (text and images) and generating detailed, explainable reports of its fact-checking process. Unlike previous approaches that were often limited to text-only analysis or lacked transparency, DEFAME integrates multiple evidence sources, including web searches, reverse image searches, and geolocation tools, to verify claims. The system was evaluated on several established benchmarks (VERITE, AVERITEC, MOCHEG) and demonstrated superior performance compared to existing methods, establishing new state-of-the-art results. Additionally, the authors created a new benchmark, CLAIMREVIEW2024+, featuring claims that occurred after the knowledge cutoff of GPT-4O to ensure more realistic evaluation scenarios. The results show that DEFAME significantly outperforms GPT-4O baselines on this new dataset, demonstrating its potential for real-time fact-checking in dynamic information environments.

**Claims And Evidence:**

The paper has several concerns regarding the experimental setup and comparisons made.
1. The results reported on the AveriTec dataset were not obtained using the default settings; instead, the paper solely relied on accuracy as the evaluation metric.
2. The comparison with GPT-4o seems unfair. A fair comparison would involve contrasting GPT-4o in a multi-turn setup with DEFAME in a multi-turn setup, as well as comparing GPT-4o in its current single-turn setup with DEFAME in a single-turn setup.
3. Lastly, the paper failed to compare its approach with existing efforts that combine Large Vision-Language Models (LVLMs) and RAG (see section Essential References Not Discussed).

**Essential References Not Discussed:**

It appears that this paper has overlooked several important related works. The paper claims that DEFAME is the first multimodal Automated Fact-Checking system capable of handling multimodal claims while also retrieving and processing multimodal evidence. However, there are many existing works that employ Large Vision-Language Models (LVLMs) with knowledge retrieval for multimodal misinformation detection.

[1] SNIFFER: Multimodal Large Language Model for Explainable Out-of-Context Misinformation Detection. CVPR 2024.
[2] MMFakeBench: A Mixed-Source Multimodal Misinformation Detection Benchmark for LVLMs. ICLR 2025 (but arXiv on 13 Jun 2024)
[3] LEMMA: Towards LVLM-Enhanced Multimodal Misinformation Detection with External Knowledge Augmentation. arXiv 2024
[4] Multimodal Misinformation Detection using Large Vision-Language Models. CIKM 2024

**Experimental Designs Or Analyses:**

See the section "Claims And Evidence" above.

**Methods And Evaluation Criteria:**

The proposed methods and evaluation criteria are well-suited for the problem of automated fact-checking, particularly in handling multimodal claims and evidence.

**Other Comments Or Suggestions:**

See above.

**Other Strengths And Weaknesses:**

1. While DEFAME represents an advancement in multimodal fact-checking, its contributions primarily lie in the workflow and prompt engineering. However, I did not see significant differences between the proposed framework and existing works that combine LVLMs and RAG. The only additions are reverse image search and geolocation, but these two actions only provide limited improvements.

2. The paper claims that DEFAME is the first multimodal Automated Fact-Checking system capable of handling multimodal claims while also retrieving and processing multimodal evidence. However, there are many existing works that employ LVLMs with knowledge retrieval for multimodal misinformation detection.

3. Additionally, many details of the proposed dataset ClaimReview 2024+ are missing. The evaluation of the proposed method is also not comprehensive, as discussed in the section "Claims and Evidence" above.

**Questions For Authors:**

No

**Relation To Broader Scientific Literature:**

In summary, DEFAME's contributions build upon and advance the state of the art in automated fact-checking。

**Theoretical Claims:**

This paper did not provide any proofs for theoretical claims.

---

> ### Author Rebuttal · Authors · 2025-03-31
>
> We value the time invested in the review and thank for the feedback and suggestions! Please find our response to your concerns below.
>
> ### Comparison to Prior Work
> Thanks for pointing out **SNIFFER**, a related work that was unintentionally omitted. Critically, unlike DEFAME, SNIFFER is **incapable of retrieving multimodal evidence** (“we input both the news caption and the text from webpages retrieved [...] into the LLM”, p. 5). It performs no planning. SNIFFER has only one tool (entity extraction) that is always executed. It cannot dynamically decide to retrieve additional evidence. SNIFFER is limited to news content and does not generalize beyond OOC detection. It is inapplicable to text-only inputs. Additionally, it requires finetuning - which DEFAME does not.
>
> **MMD-Agent** was missing from our initial submission as it was not peer-reviewed back then, but we are happy to integrate it now. MMD-Agent **does not incorporate visual evidence**. Specifically, Figure 4 and the [published code](https://github.com/liuxuannan/MMFakeBench/tree/main/eval/prompt_template/MMD_Agent) indicate that the gathered evidence is text-only. The paper only mentions Wikipedia as the source of external evidence (pp. 7 and 10). Additionally, their pipeline is static and does not allow for follow-up evidence retrieval. It does not include systematic planning. MMD-Agent is restricted to news. Finally, the outputs lack justifications and, therefore, the overall explainability of MMD-Agent is limited.
>
> You mentioned the work by Tahmasebi et al. (CIKM 2024). The method introduced there is referred to as **LVLM4F**, which our paper covers in detail, most notably in the prior work overview (Table 1) and as a baseline in Table 3. Critically, LVLM4FV **is not designed to handle multimodal claims**: “Given a **textual** claim 𝑄 and a corpus of evidences C as input [...]”, p. 3.
>
> Finally, we did not include **LEMMA** for the following reasons. First, it is not peer-reviewed. Second, LEMMA **only retrieves textual evidence**: The "Vision Evidence" in Figure 4 of the LEMMA paper refers to "a list of web page's title" (p. 6). These web pages were retrieved via reverse image search using the input image - which is probably why they call it “vision evidence.” Third, it remains unclear if LEMMA applies to text-only inputs. Finally, it cannot retrieve further evidence after the first pass, does not involve any efficient tool use planning, and does not provide any comprehensible justification generation, limiting its explainability.
>
> Considering all four references, our claim remains valid that no prior published MAFC method can handle both multimodal claims **and** multimodal evidence.
>
> ### ClaimReview2024+ Details
> Complementary to Appendix J, we are happy to add more details on CR+ in the following: Claims were collected via the Google Fact-Check API by issuing queries across ten broad topics (climate change, politics, health, …). We deduplicated results based on the review URL to avoid overlap. For each claim, we also collected the date and author (claimant) to preserve context. Label assignment was handled in two stages: Trivial labels were automatically mapped using an LLM-based script (code included in the release), while non-trivial cases were manually annotated by a PhD-level MAFC researcher. A full validation pass was conducted by a student who compared extracted content (text, label, date, claimant, image) with the original fact-check articles to ensure accuracy. Images were manually curated since the Google API only returns teaser images, which often contain overlays or composites. Manual curation ensured that claim images are as close as possible to the original ones referenced in the fact-check. Because the Google Fact-Check API aggregates claims from leading organizations such as Snopes, PolitiFact, and AFP—whose focus is on timely and harmful misinformation—we believe CR+ reflects real-world misinformation trends. We will clarify these details in the main paper for greater transparency. Thank you for the feedback.
>
> ### AVeriTeC metric
> We used accuracy as the main metric for AVeriTeC to align with recent follow-up work, where accuracy has become a common evaluation metric for veracity prediction [Singhal et al., 2024; Cao et al., 2023]. The original AVeriTeC paper conditions veracity scoring on alignment between retrieved and gold evidence, which can confound evaluation of general-purpose systems. Following subsequent work, we omit this step to enable broader comparability and support systems that retrieve evidence freely.
>
> ### Evaluation “Not Comprehensive”
> You point out that “evaluation of the proposed method is also not comprehensive” and refer to section "Claims and Evidence." There, however, you write that “the claims made in the submission are supported by clear and convincing evidence.” Please clarify.

---

> > ### Comment · Reviewer_K7Ta · 2025-04-04
> >
> > The section "claims and evidence" was updated and listed a few missing experiments. I have read the authors' responses to other reviewers that share similar concerns; some of these concerns have been addressed. Therefore, I will increase my rating to weak acceptance for this paper.

---

> > > ### Author Response · Authors · 2025-04-08
> > >
> > > We sincerely thank the reviewer for considering our paper for acceptance. We are happy to address the remaining/missed concerns in the following:
> > >
> > > ### AVeriTeC Metrics
> > > As noted in our previous response, we used accuracy as the main metric for AVeriTeC to enable comparison with recent work, where it has become a commonly reported metric for veracity evaluation. The original AVeriTeC paper conditions veracity scoring on the alignment between retrieved and gold evidence, framed as QA-style inputs. In contrast, DEFAME’s standard output is a structured fact-checking report intended to support explainability and generality across benchmarks. We implemented the required method adaptation to support the official AVeriTeC scoring protocol and included the results in Figure 5 in our paper.
> > >
> > > ### Experimental Fairness
> > > Thanks for suggesting additional experiments to increase comparison fairness. We followed your suggestion and executed GPT-4o with CoT prompting in a multi-turn variant, leveraging the same reiteration criterion as in DEFAME. The results are shown in the following table.
> > >
> > > | Method 	| Turns  | MOCHEG (F1) | VERITE (Acc.) | CR+ (acc.) |
> > > | ---------- | ------ | ----------- | ------------- | ---------- |
> > > | DEFAME 	| Multi  | **59.5**	| **84.5**  	| **68.7**   |
> > > | GPT-4o CoT | Multi  | 57.0    	| 82.3      	| 40.7   	|
> > > | DEFAME 	| Single | 47.7    	| 82.8      	| 63.3   	|
> > > | GPT-4o CoT | Single | 49.6    	| 79.7      	| 36.6   	|
> > >
> > > Even in a multi-turn setup, GPT-4o scores lower than DEFAME. However, the additional turns help GPT-4o to improve over the single-turn baseline. The multi-turn setup incentivizes GPT-4o to leverage its parametric knowledge. This is expected because most data of MOCHEG and VERITE is leaked (recall that almost all their claims are from before GPT’s knowledge cutoff). To simulate a more realistic scenario, we also evaluate on CR+ which contains mostly claims from after the knowledge cutoff. Indeed, on these “unseen” claims, GPT-4o with multi-turn lacks behind strongly, with a **gap of 28.0 percentage points** in accuracy.
> > >
> > > ### Related Work Using LVLMs and RAG
> > > We thank the reviewer for pointing at the four references—SNIFFER, MMD-Agent (MMFakeBench paper), LVLM4FV, and LEMMA—which we extensively addressed in our previous response. To complete the picture of works that combine LVLMs with RAG, to the best of our knowledge, there remains only one more method: RAGAR. Our paper already covers RAGAR in Table 1. Critically, RAGAR **cannot retrieve multimodal evidence**. (The RAGAR paper misleadingly refers to it as “multimodal evidence,” perhaps because it was retrieved with reverse image search. In fact, the evidence consists only of text or image captions.) Moreover, RAGAR reduces multimodal claims into fully verbalized descriptions. That is, all follow-up reasoning is text-only and may miss important details in the image. In contrast, DEFAME is aware of the full claim image throughout the whole pipeline, allowing it to compare it to evidence images, reason with it, etc.
> > >
> > > We are happy to add the missing references to the camera ready, complemented by the clear distinctions as pointed out in our responses.

---

### Official Review · Reviewer_qLGa · 2025-03-24

**Overall Recommendation:** 3

**Summary:**

This paper introduces DEFAME, a multimodal pipeline for open-domain text-image claim verification. DEFAME operates as a six-stage process that handles both multimodal claims and evidence while generating structured reports. It dynamically selects appropriate tools to extract and evaluate both textual and visual evidence, using web search, image search, reverse image search, and geolocation tools.
The authors evaluated DEFAME on three available benchmarks (VERITE, AVERITEC, and MOCHEG) where it surpassed previous state-of-the-art methods. They also introduced CR+, a benchmark containing claims after GPT-4o's knowledge cutoff, where DEFAME significantly outperformed GPT-4o and GPT-4o CoT. Ablation studies and human evaluations confirmed DEFAME's components each contribute to its performance and that it provides better justifications than baseline LLMs.
The authors address an important issue in the proliferation of misinformation and identify critical limitations of current MLLMs, particularly their reliance on static parametric knowledge and inability to access up-to-date evidence.

**Claims And Evidence:**

Please see other sections for my comments.

**Essential References Not Discussed:**

N/A

**Experimental Designs Or Analyses:**

1. The ablation study should be extended to include the AVeriTec and CR+ datasets.

2. The paper misses an opportunity to discuss in depth how each component of the agent system complements a baseline MLLM. While section 4.3 provides examples of how DEFAME outperforms GPT-4o, it lacks systematic analysis beyond these examples. Several questions remain unanswered: Why does Web Search significantly help MOCHEG in the ablation table? Why does VERITE benefit substantially from reverse search? The ablation should be compared alongside baseline GPTs—since GPT-4o outperforms DEFAME without Web Search, comparisons between GPT-4o + websearch and DEFAME would help determine whether the other components truly add value. Without these detailed analyses, the paper lacks convincing evidence that all components should be included.
3. The confusion matrix for MOCHEG-DEFAME in Appendix Figure 7 reveals that DEFAME struggles with classifying NEIs. The authors should confirm whether this is due to the outdated information issue mentioned in Appendix G, or provide alternative explanations.
4. The effectiveness of the planning phase requires further examination. How would performance change if all actions were taken for every example instead of being selectively planned?

**Methods And Evaluation Criteria:**

1. The paper needs a clearer explanation of the "summarize," "develop," and "justify" stages. While prompts for each stage are included in the appendix, their specific goals remain unclear. The ablation study should also evaluate performance without these steps (currently only includes w/o Develop). Consider adding ablations for w/o summarize and w/o justify as well.
2. The comparison with GPT-4o requires more methodological consistency. Currently, GPT-4o appears to be used in a single-turn manner. More meaningful comparisons would be between GPT-4o multi-turn (prompting again when the result is NEI) with DEFAME multi-turn, and between GPT-4o single-turn (current setup) with DEFAME single-turn.
3. The human evaluation section needs more detail. Information should be provided about the evaluators' backgrounds and relevant experience in fact-checking. The paper should clarify what content was shown to evaluators (claims, actions, evidence, elaboration, final judgment, justification?) and which specific components were evaluated. The claim that DEFAME "provides better justification compared to base MLLM" suggests component-specific evaluation, but it would be more reasonable to evaluate only the evaluation, judgment, and justification parts of the report.
4. In Table 3, should also include performance of other models on CR+

**Other Comments Or Suggestions:**

I think this paper has great potential, with improved discussion and a more detailed ablation (or reasonable explanation) I am willing to increase my score to accept.

**Other Strengths And Weaknesses:**

Other Strength:
1. The paper compares carefully about time and computational cost of DEFAME and different ablations
2. Appendix includes very useful information that did not make to the main paper

**Questions For Authors:**

N/A

**Relation To Broader Scientific Literature:**

Other works largely focused on isolated aspects of fact-checking—such as text-only verification, evidence retrieval, or uni-modal approaches—DEFAME integrates these perspectives into a comprehensive end-to-end solution. The authors position their work within the growing body of research on multimodal reasoning, retrieval-augmented generation, and explainable AI systems.

**Theoretical Claims:**

No proofs in this paper.

---

> ### Author Rebuttal · Authors · 2025-03-31
>
> We thank for highlighting that the “paper has great potential” and are grateful for pointing out the usefulness of the information provided in the appendix. We appreciate the time invested into the review and are happy to address the concerns in the following.
>
> ### Summarize, Develop, and Justify Stages
> - The summarize stage serves several roles. Efficiency and Comprehensiveness: Without it, retrieved web evidence (entire web articles, PDFs, etc.) would go directly into the report, resulting in an excessively long document containing unfiltered raw information, incl. ads and other irrelevant information. Report documents have length of typically a few thousand tokens, but raw evidence can go well beyond hundreds of thousands. This would increase the required computational cost by orders of magnitude and decrease the human readability of the report. Performance: Upon your request, we compared standard DEFAME against simple truncation at 1000 characters and observed a drop from 68.7% to 63.2% accuracy on CR+, confirming its practical necessity.
> - The develop stage mainly targets to determine how well the claim is supported by the gathered evidence. This is done by contrasting the claim with the evidence and performing Natural Language Inference (NLI) through Chain-of-Thought (CoT). We expect from the NLI to deduce facts helpful for the fact-check. Moreover, we expect that gaps of missing evidence become evident during that process. This stage prepares the judge stage where the actual classification happens.
> - The justify stage aims to produce a comprehensive summary of the report to serve the user as a quick explanation of the decision. It is purely explanatory and does not affect label prediction—ablating it would not change performance metrics.
>
> ### GPT-4o Comparison (Single-turn vs. Multi-turn)
> We conducted an additional experiment using multi-turn GPT-4o with CoT prompting leveraging the same reiteration criterion as in DEFAME. It achieved 40.7% accuracy on CR+, only slightly improving over the single-turn variant. The primary limitation remains GPT-4o’s lack of access to external evidence, leading to overuse of NEI.
>
> ### Human Evaluation
> As detailed in Appendix H, evaluators saw complete reports (claim, evidence, verdict, justification); baseline outputs were reformatted for fairness (Appendix K). While not trained fact-checkers, all evaluators had higher education and familiarity with MLLMs.
>
> ### Other Models on CR+
> Prior SOTA methods weren’t included on CR+ due to multiple reasons. Please refer to our response to Reviewer 3bbT (*“‘Incomplete’ Table 3”*) for more details.
>
> ### Ablation Scope
> Thank you for recommending additional ablations, which we now extended to CR+, see the results in the table below.
>
> |Variant|CR+ Acc.|
> |-|-|
> |DEFAME|**68.7**|
> |w/o Geolocation| 65.7|
> |w/o Reverse Search| 64.0|
> |w/o Image Search| 63.7|
> |w/o Web Search| 59.7|
> |Single Turn| 63.3|
> |Static Actions| *68.0*|
> |w/o Develop|67.0|
> |w/o Summarize|63.2|
> |Unimodal Develop| 65.7|
>
> The values confirm that all components contribute meaningfully to DEFAME’s performance - removing any would hurt the method. We intentionally did not extend ablations to AVeriTeC due to its unimodal nature, which renders many multimodal ablations inapplicable.
>
> ### Component Contributions
> We observe distinct tool contributions across datasets due to the fundamental differences in the tasks: VERITE targets Out Of Context (OOC) detection which (as already shown by previous work) benefits highly from applying Reverse Image Search to the input image. MOCHEG benefits from Web Search strongly as, in contrast to VERITE, it is constructed from real-world claims only.
>
> We performed an extra ablation DEFAME with GPT-4o + Web Search (GPT WS) on CR+, where GPT WS was allowed to perform a single round of evidence retrieval and to apply CoT. GPT WS achieved an accuracy of 54.7% which is clearly better than the native baselines but still strongly worse than DEFAME (accuracy 68.7%). Would you like to see additional ablations on MOCHEG and VERITE?
>
> ### NEI Confusions in MOCHEG
> Yes, many NEI errors result from outdated or time-sensitive labels, as discussed in Appendix G. Others stem from DEFAME’s current limitation of not assessing source credibility. For example, in one case, DEFAME classified the claim "Former President Jimmy Carter said 'America has no functioning democracy at this moment.’” as Supported based on an article from truthout.org. However, Snopes labeled the same claim as Not Enough Information, arguing that further verification from more authoritative or corroborated sources was necessary. This illustrates how differences in evidence sufficiency thresholds—and subjective judgments about source credibility—can lead to apparent disagreement.
>
> ### Effect of Planning
> Table 4 and the additional ablations on CR+ in the table above already include the removal of planning, referred to as “Static Actions” (all tools used every time).

---

### Decision · Program_Chairs · 2025-05-01

**Decision:**

Accept (poster)

**Comment:**

The paper proposes a multi-stage pipeline for multimodal fact-checking and a benchmark of claims from after the GPT-4 training data cutoff that are less likely to be verified through memorization. There are no lingering reviewer concerns about whether the work is technically solid, and there is a comprehensive evaluation on existing multimodal fact-checking benchmarks that indicates respectable empirical improvements for this domain. However there were concerns over lack of innovation and similarity to existing RAG-LLM solutions. In their rebuttal the authors have somewhat convincingly differentiated their work, and I do agree this paper has potential for impact. I would suggest the authors include statistical significance testing for the main results before publication.